# Understanding Fine-tuning in Approximate Unlearning: A Theoretical Perspective

**Meng Ding**
*Department of Computer Science and Engineering*
*State University of New York at Buffalo*

**Rohan Sharma**
*Department of Computer Science and Engineering*
*State University of New York at Buffalo*

**Changyou Chen**
*Department of Computer Science and Engineering*
*State University of New York at Buffalo*

**Jinhui Xu**
*School of Information Science and Technology*
*University of Science and Technology of China*

**Kaiyi Ji**
*Department of Computer Science and Engineering*
*State University of New York at Buffalo*

**Reviewed on OpenReview:** *https://openreview.net/forum?id=XXXX*

## Abstract

Machine Unlearning has emerged as a significant area of research, focusing on 'removing' specific subsets of data from a trained model. Fine-tuning (FT) methods have become one of the fundamental approaches for approximating unlearning, as they effectively retain model performance. However, it is consistently observed that naive FT methods struggle to forget the targeted data. In this paper, we present a theoretical analysis—the first to characterize FT unlearning behavior in linear models, providing a deeper exploration of this phenomenon. Our analysis reveals that while FT models can achieve zero remaining loss, they fail to forget the forgetting data, as the pretrained model retains its influence and the fine-tuning process does not adequately mitigate it. To address this, we propose a novel Retention-Based Masking (RBM) strategy that constructs a weight saliency map based on the remaining dataset, unlike existing methods that focus on the forgetting dataset. Our theoretical analysis demonstrates that RBM not only significantly improves unlearning accuracy (UA) but also ensures higher retaining accuracy (RA) by preserving overlapping features shared between the forgetting and remaining datasets. Experiments on synthetic and real-world datasets validate our theoretical insights, showing that RBM outperforms existing masking approaches in balancing UA, RA, and disparity metrics.

## 1 Introduction

Machine Unlearning has emerged as a prominent area that focuses on protecting individual privacy during the model training process, particularly adhering to legislation such as 'the right to be forgotten' (Rosen, 2011) under the General Data Protection Regulation (GDPR) (Hoofnagle et al., 2019). That is, it removes certain training samples from the trained model upon their users' data deletion request. A natural approach to machine unlearning is to retrain the model from scratch, excluding the data that needs to be forgotten;

Table 1: Cifar-10 Class-wise Forgetting Performance Comparing Retrain and Naive FT (Fine-Tuning) Method. The table compares Retrain and FT on CIFAR-10 across multiple evaluation metrics: Unlearning Accuracy (UA), Retaining Accuracy (RA), MIA-Efficacy, Test Accuracy (TA), Avg. Disparity, and Run-Time. Values in brackets indicate the gap between FT and the golden model (i.e., Retrain). Further explanations are provided in Section 5.1.

| Methods | UA | RA | Cifar-10 Class-wise Forgetting | | Avg. Disparity | Run Time |
| | | | MIA-Efficacy | TA | | |
|---|---|---|---|---|---|---|
| Retrain | $100.00_{\pm 0.00}$ | $100.00_{\pm 0.00}$ | $100.00_{\pm 0.00}$ | $94.81_{\pm 0.09}$ | 0.00 | 82.00 |
| FT | $8.36_{\pm 3.03}(91.64)$ | $40.76_{\pm 8.03}(59.24)$ | $99.92_{\pm 0.03}(0.08)$ | $94.41_{\pm 0.29}(0.40)$ | 37.84 | 2.53 |

this is known as exact unlearning. However, this method is highly computationally inefficient. To address this challenge, previous research has proposed a more relaxed definition of machine unlearning, where the unlearned model only needs to be approximately similar to one retrained from scratch. This led to the development of *approximate unlearning* methods, such as Fine-Tuning (Warnecke et al., 2021; Golatkar et al., 2020a), Gradient Ascent (Graves et al., 2021; Thudi et al., 2022), Fisher Forgetting (Becker & Liebig, 2022; Golatkar et al., 2020a), and Influence Unlearning(Izzo et al., 2021). Fine-tuning, as one of the most widely used approaches in approximate unlearning, has demonstrated its empirical effectiveness. However, it can be observed in many studies (Kurmanji et al., 2024; Warnecke et al., 2021; Golatkar et al., 2020a; Liu et al., 2024; Sharma et al., 2024) and our investigations in Table 1 that while fine-tuning may maintain the utility of the model on remaining data, it struggles to forget the targeted data. This raises a natural question:

*Why does fine-tuning fail to unlearn the forgetting data?*

To answer this question, we revisit the machine unlearning problem with a simple yet fundamental over-parameterized linear regression model and explore the behavior of fine-tuning through a theoretical perspective. Our main contributions can be summarized as follows.

- **Theoretical Understanding of Fine-Tuning.** We provide a theoretical analysis—the first to characterize FT unlearning behavior in linear models. Specifically, **1**) Based on the assumption of distinct features (Assumption 3.1), our theoretical observations, which align with empirical studies, show that the remaining loss for the fine-tuning model is zero, matching that of the golden model. Moreover, the loss of the fine-tuning model on the forgetting dataset consistently remains zero, diverging from the performance of the golden model. **2**) We extend our analysis to a more complex case when the dataset retained for model retraining shares overlapping features with the forgetting dataset. This challenges assumptions of distinct feature sets across datasets, yet the previous conclusions remain valid in this case. More discussion refers to Section 3.

- **Understanding the Benefits of Masking in Fine-Tuning.** Our analysis shows that naive fine-tuning (FT) methods fail to unlearn the forgetting data because the pretrained model retains information about this data, and the fine-tuning process does not effectively alter that retention. To address this, we propose removing the forgetting component to mitigate its retention in the pretrained model, an approach that aligns with the masking concept proposed in existing work Fan et al. (2023), which directly masks the forgetting data. However, one critical case is omitted: when the remaining data and forgetting data share similar features, it becomes unclear whether those shared features should be preserved. In Section 4, our work proves: **1**) Masking on the pretrained model can significantly improve unlearning accuracy while preserving the retaining accuracy. **2**) When considering overlapping features, retaining them does not substantially affect unlearning accuracy, but discarding them compromises the retaining accuracy.

- **Retention-Based Masking.** Building on the aforementioned analysis, we propose a novel Retention-Based Masking (RBM) strategy that constructs the weight saliency map based on the remaining dataset instead of the forgetting dataset.

  To validate our theoretical results, we conduct experiments on both synthetic and real-world datasets. First, all mask-based methods significantly improve UA compared to the naive FT method. Addi-

tionally, RBM preserves overlapping features by constructing masks based on the remaining dataset, achieving higher RA than forgetting-based methods. Furthermore, RBM consistently achieves lower average disparity, effectively balancing unlearning and retaining objectives.

## 1.1 Related Work

**Machine Unlearning Methods.** Cao & Yang (2015) first defined "Unlearning" as the removal of a sample that produces the same output on the dataset as if the sample had never been trained. The natural way to solve the problem is to retrain a model from scratch in response to each data deletion request. However, retraining is not feasible due to the limited time and constrained resources. Ginart et al. (2019) provided a relaxed definition inspired by Differential Privacy (Dwork et al., 2014), which only requires the unlearned model to produce results similar to those of retrain-from-scratch models. This led to the development of "approximate unlearning" methods, offering more efficient computational designs for machine unlearning. Guo et al. (2019); Izzo et al. (2021); Neel et al. (2021); Ullah et al. (2021); Sekhari et al. (2021) provide theoretical error guarantees by focusing on the empirical risk minimization problem under this probabilistic notion of unlearning. Golatkar et al. (2020a) proposed an information-based procedure to remove knowledge from the trained weights, without access to the original training data. Further, Golatkar et al. (2020b) approximated the weights inspired by NTK theory, addressing situations where the Hessian is not informative about where the model will converge into a null space. Mehta et al. (2022) avoid the computation of Hessian by introducing a method only computing conditional independence, which identifies the Markov Blanket of parameters requiring updates. Thudi et al. (2022) proposed a regularizer to reduce the 'verification error,' which represents the distance between the unlearned model and a retrained-from-scratch model. Kurmanji et al. (2024) bears a novel teacher-student formulation to achieve better performance towards unbounded unlearning problems. Liu et al. (2024) considers model sparsity by pruning weights before the unlearning process, thereby introducing a new unlearning paradigm. Shen et al. (2024) incorporates the variational inference and contrastive learning approaches to address the lack of supervision information (label-agnostic). Torkzadehmahani et al. (2024) leverages memorization dynamics to design localized masks that better preserve task-relevant representations, while Foster et al. (2024) computes a parameter-importance ratio between the forget and retain sets using the Fisher information matrix. In contrast, our method derives the masking strategy directly from the remaining dataset, guided by theoretical insights into feature overlap and retention.

**Machine Unlearning Theory.** For approximate unlearning, Neel et al. (2021); Thudi et al. (2022) explored algorithms for empirical risk minimization objectives, while Sekhari et al. (2021) studied population risk minimization problems, providing theoretical guarantees on both the effectiveness of unlearning and the privacy of the data subjects. Guo et al. (2019); Zhang et al. (2022) provided the certified radius with respect to data changes before and after removals, as well as the certified budget for data removals. For exact unlearning, Ullah et al. (2021) introduced the notion of algorithmic stability, called Total Variation (TV) stability, which is suited for achieving exact unlearning. This concept was further extended to the federated setting by Che et al. (2023); Tao et al. (2024). Chien et al. (2024) introduced Langevin Unlearning, which interprets noisy gradient descent as an implicit Bayesian sampling mechanism that promotes forgetting through stochastic noise injection. However, existing theoretical work has primarily focused on utility guarantees or privacy-driven dynamics, with limited analysis explaining the successes and failures of fine-tuning methods.

**Notations**: In this paper, we adhere to a consistent notation style for clarity. We use boldface lower letters such as $\mathbf{x}, \mathbf{w}$ for vectors, and boldface capital letters (e.g. $\mathbf{A}, \mathbf{H}$) for matrices. Let $\|\mathbf{A}\|_2$ denote the spectral norm of $\mathbf{A}$ and $\|\mathbf{v}\|_2$ denote the Euclidean norm of $\mathbf{v}$. For two vectors $\mathbf{u}$ and $\mathbf{v}$, their inner product is denoted by $\langle \mathbf{u}, \mathbf{v} \rangle$ or $\mathbf{u}^\top \mathbf{v}$. For two matrices $\mathbf{A}$ and $\mathbf{B}$ of appropriate dimension, their inner product is defined as $\langle \mathbf{A}, \mathbf{B} \rangle := \mathrm{tr}(\mathbf{A}^\top \mathbf{B})$. For a positive semi-definite (PSD) matrix $\mathbf{A}$ and a vector $\mathbf{v}$ of appropriate dimension, we write $\|\mathbf{v}\|_\mathbf{A}^2 := \mathbf{v}^\top \mathbf{A} \mathbf{v}$. Denote by $\mathbf{P}_m$ the projection onto the space of a matrix $\mathbf{X}_m$, i.e., $\mathbf{P}_m = \mathbf{X}_m (\mathbf{X}_m^\top \mathbf{X}_m)^{-1} \mathbf{X}_m^\top$.

## 2 Machine Unlearning in Linear Models

Let $D = \{(\mathbf{x}_i, y_i)\}_{i=1}^n$ be a training dataset consisting of $n$ data points, where $\mathbf{x}_i$ represents the feature vector, and $y_i$ is the response variable for each data point in the dataset $D$. Assume that each pair $(\mathbf{x}_i, y_i)$ is a realization of the linear regression model: $y = \mathbf{x}^\top \mathbf{w}_*$, with $\mathbf{w}_* \in \mathbb{R}^d$ being the optimal model parameter in the overparameterized regime ($n \ll d$). Machine Unlearning aims to remove (or scrub) the influence of specific training data from a trained machine learning (ML) model. Let $D_f = \{(\mathbf{x}_i, y_i)\}_{i=1}^{n_f} \subseteq D$ represents a subset whose influence we want to scrub, termed the forgetting dataset. Accordingly, the complement of $D_f$, termed the remaining dataset, is $D_r = \{(\mathbf{x}_i, y_i)\}_{i=n_f+1}^n = D \backslash D_f$. The forgetting data can be represented by stacking the feature vectors and response variables as follows:

$$\mathbf{X}_f := [\mathbf{x}_1, \mathbf{x}_2, \ldots, \mathbf{x}_{n_f}] \in \mathbb{R}^{d \times n_f}, \mathbf{y}_f := [y_1, y_2, \ldots, y_{n_f}]^\top \in \mathbb{R}^{n_f \times 1}$$

Similarly, it also holds for the remaining data:

$$\mathbf{X}_r := [\mathbf{x}_{n_f+1}, \mathbf{x}_{n_f+2}, \ldots, \mathbf{x}_n] \in \mathbb{R}^{d \times (n-n_f)}, \mathbf{y}^r := [y_{n_f+1}, y_{n_f+2}, \ldots, y_n]^\top \in \mathbb{R}^{(n-n_f) \times 1}$$

The overall dataset $\mathbf{X}$ and $\mathbf{y}$ are composed separately by concatenating $\mathbf{X}_r, \mathbf{X}_f$ and $\mathbf{y}_r, \mathbf{y}_f$.

**Learning Procedure** We consider the machine unlearning problem based on the fine-tuning method dividing the learning process into two distinct phases: Original Training and Fine-tuning (Unlearning). During the original training phase, we train a model on $n$ data points $\mathbf{X} \in \mathbb{R}^{d \times n}$ and obtain an original model $\mathbf{w}_o$ by optimizing $L(\mathbf{w}_o, D)$, where $L(\mathbf{w}, D)$ is defined as the mean-squared-error (MSE) loss: $L(\mathbf{w}, D) \triangleq \frac{1}{n} \|\mathbf{X}^\top \mathbf{w} - \mathbf{y}\|_2^2$. For the fine-tuning (unlearning) phase, we initialize with the original parameter $\mathbf{w}_o$ and proceed to retrain the model specifically on a subset of the remaining dataset $D_t \subseteq D_r$ by optimizing $L(\mathbf{w}_t, D_t)$, where $\mathbf{w}_t$ is the unlearn model by fine-tuning.

Since we work in the overparameterized regime, where $n < d$, each $\mathbf{w}$ can perfectly fit the dataset. We can express each solution $\mathbf{w}$ to the following optimization problems for Original training (OT), Unlearn via fine-tuning (FT) and Golden unlearning (GU) respectively:

$$\text{OT:} \quad \mathbf{w}_o = \operatorname*{argmin}_{\mathbf{w}} \|\mathbf{w}\|_2, \quad \text{s.t. } \mathbf{y} = \mathbf{X}^\top \mathbf{w} \tag{1}$$

$$\text{FT:} \quad \mathbf{w}_t = \operatorname*{argmin}_{\mathbf{w}} \|\mathbf{w} - \mathbf{w}_o\|_2, \quad \text{s.t. } \mathbf{y}_t = \mathbf{X}_t^\top \mathbf{w} \tag{2}$$

$$\text{GU:} \quad \mathbf{w}_g = \operatorname*{argmin}_{\mathbf{w}} \|\mathbf{w}\|_2, \quad \text{s.t. } \mathbf{y}_r = \mathbf{X}_r^\top \mathbf{w} \tag{3}$$

Our goal is to evaluate how the fine-tuning solution $\mathbf{w}_t$ differs from the golden model solution $\mathbf{w}_g$ which refers to retraining the model parameters from scratch over the remaining dataset $D_r$. Existing work has assessed machine unlearning performance from various perspectives (Graves et al., 2021; Becker & Liebig, 2022; Golatkar et al., 2020a; Song et al., 2019). In this paper, we focus particularly on the Unlearning Loss (UL) and Remaining Loss (RL), which refers to the model performance on the forgetting and remaining dataset respectively. These losses are defined as:

$$\text{RL:} \quad L(\mathbf{w}, D_r) = \frac{1}{n_r} \|\mathbf{X}_r^\top \mathbf{w} - \mathbf{y}_r\|_2^2, \quad \text{UL:} \quad L(\mathbf{w}, D_f) = \frac{1}{n_f} \|\mathbf{X}_f^\top \mathbf{w} - \mathbf{y}_f\|_2^2.$$

## 3 Naive Fine-Tuning Methods Fail To Unlearn

In empirical studies (Kurmanji et al., 2024; Warnecke et al., 2021; Golatkar et al., 2020a) and Table 1, it can be observed that fine-tuning may retain the utility of a model but struggles to forget. In this section, we revisit this phenomenon in a simplified setting, aiming to gain a basic understanding of why the vanilla fine-tuning method succeeds in retaining the model's utility on the remaining dataset but fails to forget the targeted data it was trained on.

### 3.1 Distinct Features

To simplify our analysis, we first consider distinct features with the following assumption:

**Assumption 3.1.** The datasets $\mathbf{X}_f$ and $\mathbf{X}_r$ possess distinct non-zero features, which can be denoted as $\mathbf{X}_r^\top = [\mathbf{R}^\top, \mathbf{0}]$ and $\mathbf{X}_f^\top = [\mathbf{0}, \mathbf{F}^\top]$, where $\mathbf{R} \subseteq \mathbb{R}^{d_r \times (n - n_f)}$ and $\mathbf{F} \subseteq \mathbb{R}^{d_f \times n_f}$ correspond to the non-zero parts, $d_r$ and $d_f$ are the distinct feature numbers for remaining and forgetting data, respectively, and it satisfied that $d_r + d_f = d$.

The assumption implies that each of these datasets contains features that are unique to each dataset–there is no overlap in the features present in $\mathbf{X}_f$ and $\mathbf{X}_r$. The construction of $\mathbf{X}_f$ and $\mathbf{X}_r$ can be achieved by rearranging the samples in the matrix to group non-zero features into distinct blocks. Without loss of generality, we assume a structure where each matrix contains only one zero block to clear the analysis. It can be obtained immediately from Assumption 3.1 that $\mathbf{w}_* = \mathbf{w}_*^f + \mathbf{w}_*^r$, where $\mathbf{w}_*^f$ and $\mathbf{w}_*^r$ are the optimal solution such that $\mathbf{y}^f = \mathbf{X}_f^\top \mathbf{w}_*^f$ and $\mathbf{y}^r = \mathbf{X}_r^\top \mathbf{w}_*^r$. In an ideal scenario for classification tasks, each class possesses its own unique set of features that distinctly differentiates it from other classes. We later extended our analysis to overlapping features in Section 3.2.

**Theorem 3.2.** *Suppose a model is trained by the procedure 2 and 3 separately. Under the Assumption 3.1, it holds that*

- *RL: $L(\mathbf{w}_t, D_r) = 0$, UL : $L(\mathbf{w}_t, D_f) = 0$;*

- *RL: $L(\mathbf{w}_g, D_r) = 0$, UL: $L(\mathbf{w}_g, D_f) = \|\mathbf{w}_*^f\|^2_{\frac{\mathbf{x}_f \mathbf{x}_f^\top}{n_f}}$ .*

*Here, $\mathbf{w}_t$ refers to the unlearned model via fine-tuning, $\mathbf{w}_g$ refers to the model parameter retrained from scratch, RL and UL refer to the remaining loss on the remaining data and the unlearning loss on the forgetting data.*

Theorem 3.2 presents two interesting observations: 1) The fine-tuning model can perform perfectly on the remaining dataset, which indicates that the information from training data has been preserved from the original model, $\mathbf{w}_o$, to the unlearned model via fine-tuning, $\mathbf{w}_t$. 2) The loss of the fine-tuning model on the forgetting dataset consistently remains zero, which diverges from the performance of the golden model. This suggests that the fine-tuning model is unable to forget the information it previously acquired from $\mathbf{w}_o$, which may be contradicted by catastrophic forgetting in continual learning (Parisi et al., 2019; Ding et al., 2024).

To illustrate the behavior of fine-tuning during the unlearning process more clearly, we consider the projective nature of learning. Firstly, the solution of Equation (2) can be represented as

$$\mathbf{w}_t = (\mathbf{I} - \mathbf{P}_t)\mathbf{w}_o + \mathbf{P}_t \mathbf{w}_*^r, \tag{4}$$

where $\mathbf{P}_t$ is the projection space of $\mathbf{X}_t$, the $\mathbf{I} - \mathbf{P}_t$ is the corresponding orthogonal space, and the $\mathbf{w}_o$ can be also represented $\mathbf{w}_o = \mathbf{P}\mathbf{w}_*$ with $\mathbf{P}$ being the projection space of $\mathbf{X}$. According to the property of projection Corollary C.1, multiplying any data matrix by a projection matrix preserves the components of the data that lie within the subspace defined by the projection. Moreover, under the distinct features assumption 3.1, it holds that

$$\mathbf{w}_t = \mathbf{P}\mathbf{w}_*^r + (\mathbf{P} - \mathbf{P}_t)\mathbf{w}_*^f. \tag{5}$$

Therefore, the unlearned model $\mathbf{w}_t$ from Equation (2) decomposed into two components for the unlearning process: the first part, $\mathbf{w}_*^r$, preserves the accuracy on the remaining data, while the second part, $\mathbf{w}_*^f$, also ensures accuracy on the forget data. However, the projection of $\mathbf{w}_*^f$ onto the fine-tuning space $\mathbf{P}_t$ has no effect, ultimately **resulting in the unlearned model $\mathbf{w}_t$ being exactly the same as the pretrained model $\mathbf{w}_o$**. The proof of Theorem 3.2 is provided in Appendix C.2.

### 3.2 Overlapping Features

In practical scenarios, training datasets often deviate from ideal classifications, introducing complexities such as overlapping features between subsets. This challenges assumptions of distinct feature sets across datasets. Therefore, we extend our previous analysis to address the presence of overlapped features. In the following, we begin by defining overlapped features.

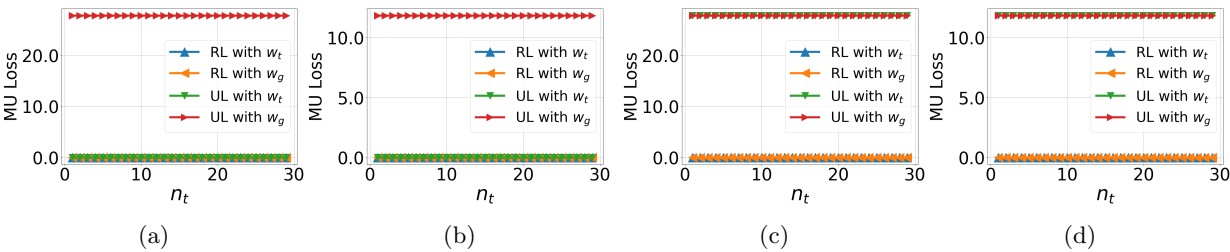

Figure 1: Machine Unlearning Performance via (Masked) Fine-tuning with (without) Overlapping Features. Section 3.1 and Figure 1b present the relationship between machine unlearning loss (i.e. RA, UA) and the number of fine-tuning data samples under distinct features and overlapping features assumptions, using naive FT method. In contrast, Figure 1c and Figure 1d show the same relationship using masked fine-tuning methods, as discussed in Section 4.

**Assumption 3.3.** The datasets $\mathbf{X}_f$ and $\mathbf{X}_r$ possess $d_r$ overlapped features, which can be structured as follows: $\mathbf{X}_r^\top = [\mathbf{R}^\top, \mathbf{L}_1^\top, \mathbf{0}]$ and $\mathbf{X}_f^\top = [\mathbf{0}, \mathbf{L}_2^\top, \mathbf{F}^\top]$, where $\mathbf{R} \subseteq \mathbb{R}^{d_r \times n_r}$ and $\mathbf{F} \subseteq \mathbb{R}^{d_f \times n_f}$ represent the distinct features of the remaining and forgetting data, respectively. $\mathbf{L}_1 \subseteq \mathbb{R}^{d_{lap} \times n_r}$ and $\mathbf{L}_2 \subseteq \mathbb{R}^{d_{lap} \times n_f}$ denote the overlapped parts.

Similarly to Assumption 3.1, $d_r$ and $d_f$ are the distinct feature numbers for remaining and forgetting data, while the equation $d_r + d_{lap} + d_f = d$ holds. It can also indicates that the optimal solution can be decomposed into $\mathbf{w}_* = \mathbf{w}_*^f + \mathbf{w}_*^{lap} + \mathbf{w}_*^r$ such that $\mathbf{y}^f = \mathbf{X}_f^\top(\mathbf{w}_*^f + \mathbf{w}_*^{lap})$ and $\mathbf{y}^r = \mathbf{X}_r^\top(\mathbf{w}_*^r + \mathbf{w}_*^{lap})$.

**Theorem 3.4.** *Suppose a model is trained by the procedure 2 and 3 separately. Under the Assumption 3.3, it holds that*

- *RL: $L(\mathbf{w}_t, D_r) = 0$, UL: $L(\mathbf{w}_t, D_f) = 0$;*

- *RL: $L(\mathbf{w}_g, D_r) = 0$, UL: $L(\mathbf{w}_g, D_f) = \|\mathbf{P}_r\mathbf{w}_*^r + \mathbf{P}_r\mathbf{w}_*^{lap} - (\mathbf{w}_*^f + \mathbf{w}_*^{lap})\|^2_{\frac{1}{n_f}\mathbf{X}_f\mathbf{X}_f^\top}$.*

Theorem 3.4 shows that the previous conclusions remain valid under the assumptions of overlapping features, as the information from all training data, including forget data, is preserved from the pretrained model, $\mathbf{w}_o$, to the unlearned model through fine-tuning, $\mathbf{w}_t$. Consequently, the loss on both the remaining dataset and the forgetting dataset for the fine-tuning model is zero. Additionally, an interesting observation is that the number of overlapping features does not impact the unlearning accuracy of the fine-tuning model. The proof of Theorem 3.4 is provided in Appendix C.3.

Both Theorem 3.2 and Theorem 3.4 present similar findings regarding the performance of the unlearned model through fine-tuning. We run a synthetic experiment to validate these results (more experimental details in Appendix B). In Section 3.1 and Figure 1b, both distinct and overlapping feature assumptions demonstrate the same results: 1) The remaining loss of fine-tuning model $\mathbf{w}_t$ and golden model $\mathbf{w}_g$ is zero, indicating that the fine-tuning model performs equivalently to the golden model, successfully retaining the model's utility on the remaining dataset. 2) The unlearning loss of the fine-tuning model consistently remains at zero, differing from the golden model, suggesting that the fine-tuning model fails to forget the information obtained from the pretrained model. These empirical findings align well with our theoretical analysis.

## 4 Eliminating Forgetting Data Features from Pre-Trained Model Enhances Unlearning

Compared to the golden model $\mathbf{w}_g = \mathbf{P}_r\mathbf{w}_*^r$, the unlearned model can be viewed as having an additional second term as

$$\mathbf{w}_t = \mathbf{P}\mathbf{w}_*^r + (\mathbf{P} - \mathbf{P}_t)\mathbf{w}_*^f.$$

This additional term $(\mathbf{P} - \mathbf{P}_t)\mathbf{w}_*^f$ represents the residual influence of the data intended to be forgotten on the unlearned model, contributing to the unlearning accuracy (UA) gap between $\mathbf{w}_t$ and the golden model

$\mathbf{w}_g$. A natural approach to mitigate this gap might involve making the fine-tuning space converge toward the pretraining space-that is, aligning $\mathbf{P}_t$ with $\mathbf{P}_r$. However, this strategy is inefficient and contradictory, as it would lead to the optimal solution for the fine-tuning dataset becoming identical to that of the entire dataset, undermining the purpose and benefits of fine-tuning.

Inspired by the formulation of the unlearned model:

$$\mathbf{w}_t = (\mathbf{I} - \mathbf{P}_t)\mathbf{w}_o + \mathbf{P}_t\mathbf{w}_*^r.$$

To mitigate the UA gap between the fine-tuning model and the golden model, it becomes evident that the remaining portion of the pretrained model does not contribute to UA. Specifically, the components of the pretrained model $\mathbf{w}_o$ associated with the forgetting data ($\mathbf{w}_*^f$) do not enhance performance on the remaining dataset $D_r$. Therefore, if we can eliminate the forgetting component—specifically by removing the $\mathbf{w}_*^f$ term from $\mathbf{w}_o$—the divergence can be addressed. In the following, we provide a formal description of this modification. Consider the same learning procedure Equation (1) to obtain the pretrained model $\mathbf{w}_o$. Prior to unlearning through fine-tuning, we modify $\mathbf{w}_o$ by removing components associated with the forgetting data. Specifically, we construct a modified model $\hat{\mathbf{w}}_o$ as follows:

**1. Distinct Features Scenario.** When the features of $D_r$ and $D_f$ are distinct, we construct $\hat{\mathbf{w}}_o$ by retaining only the components corresponding to $D_r$ and setting the rest to zero. Formally, we define $\hat{\mathbf{w}}_o$ as $\hat{\mathbf{w}}_o'[0:d_r] = \mathbf{w}_o[0:d_r]$ or equivalently can be understood as:

$$\hat{\mathbf{w}}_o[i] = \begin{cases} \mathbf{w}_o[i], & \text{if } i \in \text{features of } D_r, \\ 0, & \text{otherwise.} \end{cases}$$

**2. Overlapping Features Scenario.** When features overlap across $D_r$ and $D_f$, we consider two cases:

- **Option A** (Retaining Overlapping Features): We retain the overlapping features between $D_r$ and $D_f$, which can be expressed as $\hat{\mathbf{w}}_o[0:d_r + d_{lap}] = \mathbf{w}_o[0:d_r + d_{lap}]$ or equivalently

$$\hat{\mathbf{w}}_o[i] = \begin{cases} \mathbf{w}_o[i], & \text{if } i \in \text{features of} \\ & \quad D_r \cup \text{overlapping features}, \\ 0, & \text{otherwise.} \end{cases}$$

- **Option B** (Discarding Overlapping Features): We discard the overlapping features, which can be expressed as $\hat{\mathbf{w}}_o'[0:d_r] = \mathbf{w}_o[0:d_r]$ or equivalently

$$\hat{\mathbf{w}}_o'[i] = \begin{cases} \mathbf{w}_o[i], & \text{if } i \in \text{features of } D_r, \\ 0, & \text{otherwise.} \end{cases}$$

**Theorem 4.1.** *Let $\mathbf{w}_o$ be a pretrained model obtained the overall dataset $D = D_r \cup D_f$. Before performing unlearning (fine-tuning), we modify $\mathbf{w}_o$ to remove the components associated with $D_f$ as described above. Then, using the modified models $\hat{\mathbf{w}}_o$ ($\hat{\mathbf{w}}_o'$) in the unlearning process yields*

1. ***In the distinct features scenario*** *(i.e., under the Assumption 3.1), we have:*
   *RL: $L(\hat{\mathbf{w}}_t, D_r) = 0$; UL: $L(\hat{\mathbf{w}}_t, D_f) = \|\mathbf{w}_*^f\|^2_{\frac{\mathbf{X}_f\mathbf{X}_f^\top}{n_f}}$,*

2. ***In the overlapping features scenario*** *(i.e., under Assumption 3.3), we have:*
   - ***Option A*** *(Retaining Overlapping Features):*
     *RL: $L(\hat{\mathbf{w}}_t, D_r) = 0$;*
     *UL: $L(\hat{\mathbf{w}}_t, D_f) = \|\mathbf{P}\mathbf{w}_*^r + \mathbf{P}\mathbf{w}_*^{lap} - (\mathbf{w}_*^f + \mathbf{w}_*^{lap})\|^2_{\frac{1}{n_f}\mathbf{X}_f\mathbf{X}_f^\top}$.*
   - ***Option B*** *(Discarding Overlapping Features):*
     *RL: $L(\hat{\mathbf{w}}_t', D_r) = \|(\mathbf{I} - \mathbf{P}_t)\mathbf{w}_*^{lap}\|^2_{\frac{1}{n_r}\mathbf{X}_r\mathbf{X}_r^\top}$*
     *UL: $L(\hat{\mathbf{w}}_t', D_f) = \|\mathbf{P}\mathbf{w}_*^r + \mathbf{P}_t\mathbf{w}_*^{lap} - (\mathbf{w}_*^f + \mathbf{w}_*^{lap})\|^2_{\frac{1}{n_f}\mathbf{X}_f\mathbf{X}_f^\top}$.*

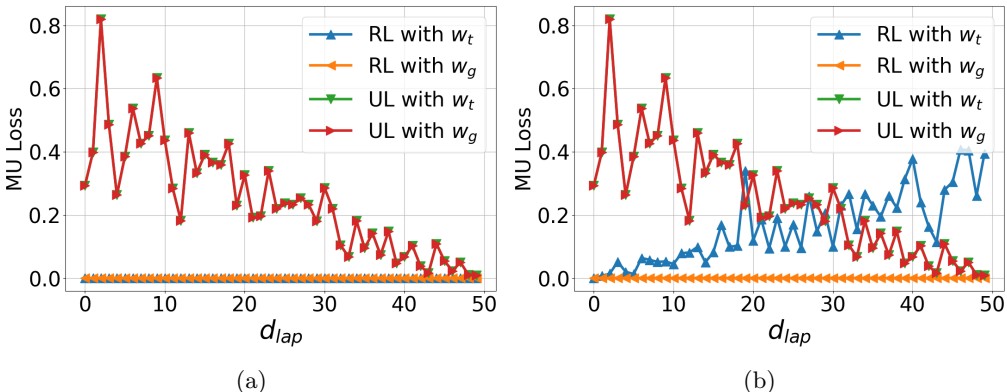

(a)

(b)

Figure 2: Comparison of Machine Unlearning Loss with and without Overlapping Features. Figure 2a retains overlapping features from the pretrained model, showing the matching performance between masked $\mathbf{w}_t$ model and golden model $\mathbf{w}_g$; Figure 2b discards the overlapping features, showing a decline in retaining accuracy.

According to Theorem 4.1, under the distinct features assumption, the masked unlearned model achieves the same remaining and unlearning loss as the golden model. Furthermore, when considering overlapping features, if the overlapped component from the pretrained model is retained, the remaining loss remains zero, as with the golden model, while the unlearning loss differs only in the projection component. This difference can be considered negligible when applied to $\mathbf{w}_*^r$ and $\mathbf{w}_*^{lap}$ due to the model assumption. Figure 1c and Figure 1d verify our theoretical conclusions. However, if the overlapped component is discarded from the pretrained model, the remaining loss is no longer zero, and there is a small change to the unlearning loss that can be overlooked. These findings offer several insights into the design of machine unlearning algorithms: **1) Masking on the pretrained model can significantly improve unlearning accuracy while preserving the retaining accuracy**. If we can identify the component of the pretrained model related to the forgetting data, applying masking to this component can further enhance UA. Our theorem also explains recent related works, such as Liu et al. (2024); Fan et al. (2023), which apply a mask to the pretrained model either randomly or by regularizing the weights associated with the forgetting data to provide better unlearn performance. These methods share the same underlying principle discussed here. However, one overlook a critical scenario: when the remaining data and forgetting data share similar features, it becomes unclear whether those shared features should be retained. As shown in Theorem 4.1, our work provides a clear resolution to this question. **2) When considering overlapping features, retaining them does not substantially affect unlearning accuracy, but discarding them compromises the retaining accuracy**. As shown in Theorem 4.1, the remaining loss can not retain zero unless the remaining data $\mathbf{X}_r$ can be fully represented by the fine-tuning space, meaning $\mathbf{P}_t\mathbf{X}_r = \mathbf{X}_r$. Additionally, as the number of overlapping features increases, the impact on both remaining and unlearning loss becomes more significant. Discarding too many overlapping components can lead to instability in the retaining accuracy, as the model loses essential information needed to represent $D_r$, which in turn causes the remaining loss to increase. Figure 1 and Figure 2 validate our theoretical findings. The proof of Theorem 4.1 is provided in Appendix C.4.

## 5 Rethinking Masking in Approximate Unlearning

### 5.1 Discarding Overlapping Features May Harm Retaining Accuracy

In Section 4 we show that once the components of the pretrained model related to the forgetting data are identified and removed, unlearning accuracy can be significantly improved. In practice, an existing work Fan et al. (2023) shares the same principles as discussed above. Specifically, it constructs the desired weight saliency map by leveraging the gradient of a loss function with respect to the model weights on the forgetting dataset:

$$\mathbf{m}_f = \mathbb{1}\left(\left|\nabla_{\mathbf{w}}L\left(\mathbf{w}; D_f\right)\right|_{\mathbf{w}=\mathbf{w}_o} \geq \gamma\right), \tag{6}$$

where $\mathbb{1}(g \geq \gamma)$ is an element-wise indicator function that outputs 1 for the $i$-th element if $g_i \geq \gamma$ and 0 otherwise, $|\cdot|$ denotes the element-wise absolute value operation, and $\gamma > 0$ is a hard threshold. Variables with larger gradients, which are highly associated with the forgetting dataset, are identified and subsequently masked. Then, the unlearn model can be updated by:

$$\mathbf{w}_{u_f} = \underbrace{\mathbf{m}_f \odot (\Delta\mathbf{w} + \mathbf{w}_o)}_{\text{salient weights}} + \underbrace{(\mathbf{1} - \mathbf{m}_f) \odot \mathbf{w}_o}_{\text{original weights}}, \tag{7}$$

where $\mathbf{w}_o$ represents the original model, $\mathbf{1}$ denotes an all-one vector and $\Delta\mathbf{w}$ is optimized by the following problem:

$$\min_{\Delta\mathbf{w}} \mathcal{L}(\mathbf{w}_{u_f}) := \mathbb{E}_{(\mathbf{x},y)\sim D_f, y'\neq y}[\ell_{\text{CE}}(\mathbf{w}_{u_f}; \mathbf{x}, y')] + \alpha\mathbb{E}_{(\mathbf{x},y)\sim D_r}[\ell_{\text{CE}}(\mathbf{w}_{u_f}; \mathbf{x}, y)], \tag{8}$$

where $y'$ is the random label of the image $\mathbf{x}$ different from $y$, $\alpha$ is a regularization parameter and $\ell_{\text{CE}}$ is the cross-entropy (CE) loss for supervised classification.

However, Equation (7) overlooks the scenario where the remaining data and the forgetting data share similar model weights, making it unclear whether those weights should be preserved. In Theorem 4.1, we demonstrate that when considering overlapping features (features shared between the remaining dataset and the forgetting dataset), retaining them is more beneficial than discarding them, which contrasts slightly with the approach in Fan et al. (2023). Specifically, we show that discarding overlapping features compromises retaining accuracy while retaining them ensures better performance for the remaining dataset.

Based on this analysis, we propose an alternative principle–Retention-Based Masking: the weight saliency map should be constructed based on the remaining dataset rather than the forgetting dataset:

$$\mathbf{1} - \mathbf{m}_r = \mathbf{1} - \mathbb{1}\left(\left|\nabla_\mathbf{w}L(\mathbf{w}; D_r)\right|_{\mathbf{w}=\mathbf{w}_o} \geq \gamma\right). \tag{9}$$

Therefore, the unlearn model will be updated by

$$\mathbf{w}_{u_r} = \underbrace{(\mathbf{1} - \mathbf{m}_r) \odot (\Delta\mathbf{w} + \mathbf{w}_o)}_{\text{salient weights}} + \underbrace{\mathbf{m}_r \odot \mathbf{w}_o}_{\text{original weights}}. \tag{10}$$

Here, $\Delta\mathbf{w}$ is optimized using Equation (8) with the retention-based mask, which combines the forgetting loss on the random labels of forgotten data with the retaining loss on the remaining data.

## 5.2 Experiment

In the following, we verify our theoretical insights by evaluating the effectiveness of the mask-based FT machine unlearning methods through numerical experiments.

**Experimental Setup.** We conduct our evaluations using the ResNet-18 backbone across the methods. The network is initially trained for classification over the CIFAR datasets for 182 epochs with an initial learning rate of 0.1 following a cosine decay schedule. For the unlearning procedure, the learning rate is set to 0.02 for our method and 0.013 for the SalUn method, following the recommendations from the official repository. We set the number of unlearning epochs to 10. However, we observe that both methods benefit from a reduced number of unlearning epochs (5) in the random forgetting scenario on CIFAR-100. We set the sparsity at 50%. Unlearning using the fine-tuning method employs a learning rate of 0.01 for 10 epochs. All methods utilize the SGD optimizer.

**Datasets and Models.** Our baseline methods include the naive fine-tuning approach (Golatkar et al., 2020a; Warnecke et al., 2021) and SalUn (Fan et al., 2023), both implemented on ResNet-18 (He et al., 2016). For comparison, we also include the golden retrained model as a reference. Our experiments will focus on image classification using the CIFAR-10 (Krizhevsky et al., 2009) and CIFAR-100 (Krizhevsky et al., 2009). More details on the experimental setup will be provided in Appendix B.

**Evaluation Metrics.** We follow the existing work to assess machine unlearning performance from different aspects (Golatkar et al., 2020a; Graves et al., 2021; Thudi et al., 2022; Liu et al., 2024; Sharma et al., 2024; Zhu et al., 2024). Specifically, we focus on the following evaluation metrics:

Table 2: **Cifar-10, Cifar-100 Comparison.** We evaluate the unlearning performance on Cifar datasets using Dense models only. Evaluation metrics from Section 5.1 are employed. Metrics are reported as mean $\pm$ standard deviation across five seeds. (**UA:** Unlearning Acc., **RA:** Retaining Acc., **TA:** Test Acc.)

| Methods | UA | MIA-Efficacy | RA | TA | Avg. Disparity | Run Time (min) |
|---|---|---|---|---|---|---|
| | | | Cifar-10 Random Forgetting | | | |
| Retrain | $5.80_{\pm 0.12}$ | $13.91_{\pm 0.15}$ | $100.00_{\pm 0.00}$ | $94.30_{\pm 0.13}$ | 0.00 | 82.15 |
| FT | $0.18_{\pm 0.04}(5.62)$ | $1.70_{\pm 0.10}(12.21)$ | $99.92_{\pm 0.07}(0.08)$ | $94.25_{\pm 0.15}(0.05)$ | 4.48 | 2.51 |
| SalUn | $2.38_{\pm 0.31}(3.42)$ | $15.45_{\pm 0.14}(1.54)$ | $99.62_{\pm 0.22}(0.38)$ | $94.11_{\pm 0.23}(0.19)$ | 1.39 | **2.50** |
| **Ours** | $3.24_{\pm 0.12}(\mathbf{2.56})$ | $14.22_{\pm 0.11}(\mathbf{0.31})$ | $99.72_{\pm 0.20}(0.28)$ | $93.90_{\pm 0.10}(0.40)$ | **0.88** | **2.50** |
| | | | Cifar-10 Class-wise Forgetting | | | |
| Retrain | $100.00_{\pm 0.00}$ | $100.00_{\pm 0.00}$ | $100.00_{\pm 0.00}$ | $94.81_{\pm 0.09}$ | 0.00 | 82.00 |
| FT | $8.36_{\pm 3.03}(91.64)$ | $40.76_{\pm 8.03}(59.24)$ | $99.92_{\pm 0.03}(\mathbf{0.08})$ | $94.41_{\pm 0.29}(0.40)$ | 37.84 | 2.50 |
| SalUn | $99.78_{\pm 0.15}(0.22)$ | $100.00_{\pm 0.00}(\mathbf{0.00})$ | $99.47_{\pm 0.23}(0.53)$ | $93.55_{\pm 0.44}(1.26)$ | 0.50 | **2.50** |
| **Ours** | $99.98_{\pm 0.02}(\mathbf{0.02})$ | $100.00_{\pm 0.00}(\mathbf{0.00})$ | $99.69_{\pm 0.30}(0.31)$ | $93.44_{\pm 0.30}(1.37)$ | **0.42** | **2.50** |
| | | | Cifar-100 Random Forgetting | | | |
| Retrain | $24.75_{\pm 0.11}$ | $49.68_{\pm 0.35}$ | $99.98_{\pm 0.01}$ | $74.57_{\pm 0.06}$ | 0.00 | 82.33 |
| FT | $0.11_{\pm 0.03}(24.64)$ | $5.66_{\pm 0.47}(44.02)$ | $99.97_{\pm 0.01}(\mathbf{0.01})$ | $75.45_{\pm 0.17}(\mathbf{0.88})$ | 16.65 | 2.50 |
| SalUn | $27.75_{\pm 0.33}(3.00)$ | $71.42_{\pm 0.11}(21.74)$ | $98.65_{\pm 0.29}(1.33)$ | $68.97_{\pm 0.39}(5.60)$ | 7.91 | **1.3** |
| **Ours** | $23.53_{\pm 0.22}(\mathbf{1.22})$ | $69.02_{\pm 0.09}(\mathbf{19.34})$ | $98.74_{\pm 0.12}(1.24)$ | $69.16_{\pm 0.11}(5.41)$ | **6.80** | **1.3** |
| | | | Cifar-100 Class-wise Forgetting | | | |
| Retrain | $100.00_{\pm 0.00}$ | $100.00_{\pm 0.00}$ | $99.98_{\pm 0.01}$ | $73.75_{\pm 0.20}$ | 0.00 | 82.22 |
| FT | $11.55_{\pm 7.1}(88.45)$ | $59.33_{\pm 17.57}(40.67)$ | $99.78_{\pm 0.03}(0.20)$ | $74.61_{\pm 0.30}(0.86)$ | 32.55 | 2.52 |
| SalUn | $100.00_{\pm 0.00}(\mathbf{0.00})$ | $100.00_{\pm 0.00}(\mathbf{0.00})$ | $99.63_{\pm 0.10}(0.35)$ | $74.76_{\pm 0.31}(1.01)$ | 0.35 | **2.50** |
| **Ours** | $100.00_{\pm 0.00}(\mathbf{0.00})$ | $100.00_{\pm 0.00}(\mathbf{0.00})$ | $99.90_{\pm 0.06}(\mathbf{0.08})$ | $74.47_{\pm 0.72}(\mathbf{0.72})$ | **0.19** | **2.50** |

- Unlearning accuracy (UA): We define $\mathrm{UA}(\mathbf{w}_t) = 1 - \mathrm{Acc}_{D_f}(\mathbf{w}_t)$ as Liu et al. (2024), measuring how effectively the model has forgotten the targeted data. Here $\mathrm{Acc}_{D_f}(\mathbf{w}_t)$ is the accuracy of the unlearned model on the forgetting dataset.

- Membership inference attack (MIA) on $D_f$ (MIA-Efficacy): The efficacy of MIA on the forget dataset, which assesses whether the model still retains any identifiable information about the forgetting data.

- Retaining accuracy (RA): The accuracy of the model on the remaining dataset $D_r$ after unlearning, measuring how well the model retains its performance from the pretrained model.

- Testing accuracy (TA): The accuracy of the model on the independent test dataset, indicating its generalization ability after unlearning.

- Average Disparity (Avg. Disparity): The mean absolute difference between the performance metrics of the unlearned model and the retrained model, calculated across all evaluation aspects.

- Run-time efficiency (RTE): RTE evaluates the computational efficiency of the unlearning process, including the run-time cost taken to execute the unlearning procedure.

Note that a smaller performance gap between the unlearned model and the golden retrained model indicates the better performance of approximate unlearning.

**Masking Strategies Enhance Unlearning Accuracy in Naive Fine-Tuning.** The performance of Retention-Based Masking (Ours) and Forgetting-Based Masking (SalUn) against Fine-Tuning highlights how masking significantly improves unlearning accuracy, measured as the distance from the retrained model. Specifically, in CIFAR-10 Class-wise Forgetting, naive fine-tuning exhibits a significant gap of 91.64 compared to the retrained model, which is drastically reduced by SalUn (0.22) and further refined by RBM to 0.02. Similarly, for CIFAR-100 Class-wise Forgetting, fine-tuning shows a large gap of 87.96, reduced to 0.00 by SalUn and maintained by RBM at 0.00. In random forgetting scenarios, in CIFAR-10 Random Forgetting, fine-tuning achieves a gap of 5.62, while SalUn reduces this to 3.42 and RBM further minimizes it to 2.56. A similar trend is observed for CIFAR-100 Random Forgetting, where fine-tuning starts with a gap of 24.64,

SalUn reduces it to 3.00, and RBM achieves a further improvement to 1.22. These results demonstrate that masking strategies, whether leveraging forgetting-based or retention-based saliency, are essential for enhancing unlearning accuracy in naive fine-tuning.

**Retention-Based Masking Improves Retaining Accuracy.** Next, we compare retention-based masking (Ours) and forgetting-based masking (SalUn) to evaluate how different masking approaches impact retaining accuracy, measured by the distance from the retrained model. For CIFAR-10 Random Forgetting, SalUn achieves a gap of 0.38, while RBM improves it to 0.28. For CIFAR-100 Random Forgetting, SalUn shows a gap of 1.33, whereas RBM significantly reduces it to 1.24. In class-wise forgetting scenarios, RBM consistently maintains an advantage: for CIFAR-10 Classwise Forgetting, SalUn achieves a gap of 0.53 compared to 0.31 for RBM. Similarly, for CIFAR-100 Class-wise Forgetting, SalUn achieves a gap of 0.35, while RBM further improves this to 0.08. These results underscore that the retention-based saliency approach is more effective in preserving the performance of the remaining dataset, aligning with our theoretical insights in Theorem 4.1.

**Retention-Based Masking Minimizes Average Disparity Across Tasks.** In addition to improving retaining accuracy, RBM also reduces average disparity across all tasks. For CIFAR-10 Random Forgetting, RBM reduces the disparity to 0.88, compared to 1.39 for SalUn. Similarly, in CIFAR-100 Random Forgetting, RBM achieves a reduced disparity of 6.80, lower than 7.91 for SalUn. In CIFAR-10 Classwise Forgetting, RBM reduces the disparity to 0.42, compared to 0.50 for SalUn. For CIFAR-100 Class-wise Forgetting, RBM achieves an exceptionally low disparity of 0.19, significantly lower than 0.35 for SalUn. These results further validate the robustness of RBM in effectively balancing retaining and unlearning objectives.

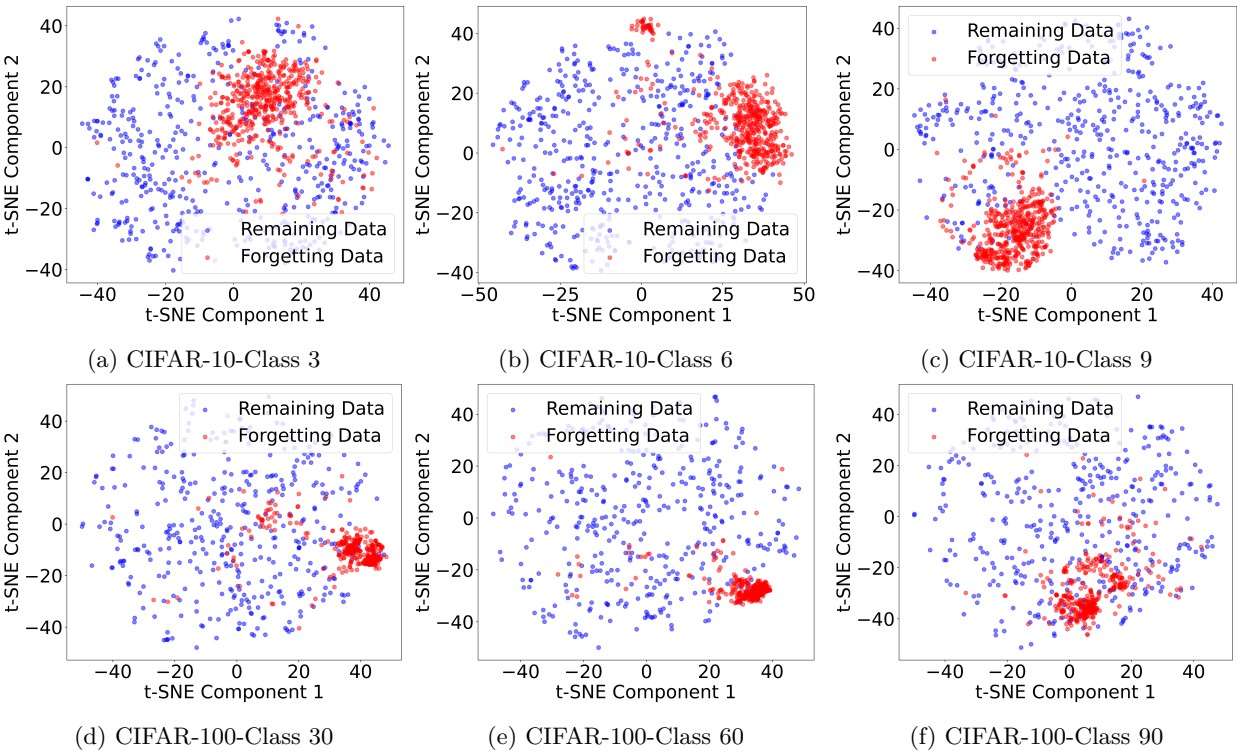

Figure 3: Visualization of Remaining Data and Forgetting Data Features Across Various Dataset. Figures 3a-3c focus on classes 3, 6, and 9 in CIFAR-10 and Figures 3d-3f focus on classes 30, 60, and 90 in CIFAR-100.

**Visualization of Remaining Data and Forgetting Data Features.** The visualization in Figure 3 uses t-SNE to project feature representations of the forgetting and remaining data in CIFAR-10 and CIFAR-100 datasets. The red points correspond to forgetting data, and the blue points represent remaining data. This visualization aims to demonstrate that, in class-wise datasets, the unlearning task for a specific class may involve distinct features. In such cases, naive fine-tuning (FT) methods tend to contribute less towards forgetting the class and focus more on retaining features from the pretrained model.

## 6 Conclusion

In conclusion, we present the first theoretical analysis of fine-tuning methods for machine unlearning within a linear regression framework. Our analysis, covering two scenarios—distinct and overlapping feature sets—demonstrates that while fine-tuning can achieve optimal retaining accuracy, it fails to fully unlearn the forgetting dataset. This failure arises from the pretrained model retaining information about the forgetting data. To address this, we propose a theoretical solution and introduce Retention-Based Masking (RBM), a strategy that constructs masks based on the remaining dataset to preserve overlapping features. Our experimental results demonstrate that RBM achieves a better balance between unlearning and retaining objectives compared to forgotten-based masking approaches.

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

## A  Discussion

Our analysis is limited to the overparameterized linear regression setting, which simplifies understanding but does not fully capture the complexity of modern nonlinear models. The assumptions of distinct and overlapping features are idealized and may not directly reflect real-world data distributions. Empirically, our evaluation is restricted to moderate-scale datasets and fine-tuning–based baselines, leaving broader comparisons to future work. Extending our framework to nonlinear or large-scale settings and exploring efficient or certified unlearning mechanisms are promising directions for future research.

## B  Experimental Details

### B.1  Verification via Simulation

To empirically validate the theoretical findings presented in Theorem 3.2, Theorem 3.4 and Theorem 4.1 regarding the performance of unlearned models through fine-tuning, we first conducted a series of synthetic experiments.

**Data Generation.** We constructed two data matrices, $\mathbf{X}_r$ and $\mathbf{X}_f$, representing the remaining data and the forgetting data, respectively. The remaining data matrix, $\mathbf{X}_r^\top$, was structured as $[\mathbf{R}^\top, \mathbf{L}_1^\top, \mathbf{0}]$, and the forgetting data matrix, $\mathbf{X}_f^\top$, as $[\mathbf{0}, \mathbf{L}_2^\top, \mathbf{F}^\top]$, where $\mathbf{L}_1^\top = \mathbf{L}_2^\top = \mathbf{0}$ to enforce the non-overlapping case. Here, $\mathbf{R}^\top$ and $\mathbf{F}^\top$ are random matrices corresponding to different feature sets, and the zeros represent the distinct features across the datasets. We set the total number of data points to $n = 40$ and the total number of features to $d = 40$. The remaining data consisted of $n_r = 30$ samples with $d_r = 20$ features, while the forgetting data comprised $n_f = 10$ samples with $d_f = d - d_r = 20$ features. To simulate a controlled environment, we fixed the number of overlapping features to $d_{lap} = 0$ and $d_{lap} = 8$ for non-overlapping case and overlapping case, respectively.

**Label Generation.** We generated the true coefficient vector $\mathbf{w}_* \in \mathbb{R}^d$ by sampling from a standard normal distribution. The labels were created using a linear regression model without added noise: $\mathbf{y} = \mathbf{X}^\top \mathbf{w}_*$. The labels were partitioned into $\mathbf{y}_r$ and $\mathbf{y}_f$, corresponding to the remaining and forgetting data, respectively.

**Model Training.** To compare the effects of fine-tuning, we considered two models: the fine-tuning model $\mathbf{w}_t$ and the golden model $\mathbf{w}_g$. Specifically, $\mathbf{w}_t$ was obtained by fine-tuning on a subset of the remaining data, denoted as $\mathbf{X}_t$, which consisted of the first $n_t$ data points from $\mathbf{X}_r$. The value of $n_t$ varied from 1 to $n_r - 1$ to study the impact of the fine-tuning data size. The initial model $\mathbf{w}_o$ was derived from the entire dataset $\mathbf{X}$ and calculated by the Equation (1). $\mathbf{w}_g$ was trained from scratch on the entire remaining data $\mathbf{X}_r$ and computed by solving Equation (2). If considering the masked case in synthetic data, the masked pretrained model will be constructed by zeroing out the coefficients corresponding to the forgetting data features with (without) overlapping features.

**Evaluation Metrics.** The performance of the models was assessed using the Mean Squared Error (MSE) on both the remaining and forgetting data:

- Remaining Data Loss (RA Loss): $\mathrm{MSE}_{\mathrm{RA}}(\mathbf{w}) = \frac{1}{n_r}\|\mathbf{X}_r\mathbf{w} - \mathbf{y}_r\|^2$

- Unlearning Data Loss (UA Loss): $\mathrm{MSE}_{\mathrm{UA}}(\mathbf{w}) = \frac{1}{n_f}\|\mathbf{X}_f\mathbf{w} - \mathbf{y}_f\|^2$.

**Experimental Results** Figure 1c and Figure 1d illustrate that the masked fine-tuning method discussed in Section 4 can significantly improve unlearning accuracy while preserving the retaining accuracy. Specifically, both the remaining loss and unlearning loss of $\hat{\mathbf{w}}_t$ perfectly match those of the golden model under both distinct and overlapping feature scenarios. Additionally, Figure 2 present comparisons of machine unlearning loss for different approaches to handling overlapping features: Figure 2a retains overlapping features from the pretrained model, demonstrating matching performance between the masked $\mathbf{w}_t$ model and golden model $\mathbf{w}_g$; whereas Figure 2b discards the overlapping features, resulting in a decline in retaining accuracy. These empirical results align well with our theoretical findings.

### B.2 Additional Real-world Details

Table 3: **Unlearning Results on TinyImageNet.** Performance comparison on the more challenging TinyImageNet dataset. Random 10% forgetting scenario with 50% masking. (**RA:** Retaining Accuracy, **UA:** Unlearning Accuracy, **TA:** Test Accuracy, **MIA Efficacy:** Membership Inference Attack Efficacy, **Avg. Disparity:** utility–privacy imbalance).

| Method | RA | UA | TA | MIA Efficacy | Avg. Disparity |
|---|---|---|---|---|---|
| Retrain | 99.98 | 39.91 | 62.21 | 65.13 | — |
| RBM | 96.89 | 47.23 | 59.91 | 69.98 | 4.39 |
| SalUn | 97.20 | 31.30 | 59.10 | 71.08 | 5.11 |

We further evaluate our approach on the TinyImageNet dataset. This dataset is a subset of ImageNet, containing 200 classes with 500 training samples per class and 50 samples each for validation and testing. We consider a random forgetting scenario involving 10% of the data, with a masking ratio of 50% for both methods. The larger class space and higher visual diversity make this a particularly challenging benchmark. Nevertheless, our approach substantially outperforms the baseline method, showing only a moderate decline in test accuracy—reflecting the dataset's inherent complexity. RBM maintains high retention accuracy while effectively mitigating MIA attacks to levels comparable to a retrained model, thereby achieving a lower average disparity relative to this standard.

Table 4: **SVHN Unlearning Results using ViT Tiny.** Evaluation of unlearning methods on SVHN. **RA:** Retaining Accuracy, **UA:** Unlearning Accuracy, **TA:** Test Accuracy, **MIA Efficacy:** Membership Inference Attack Efficacy.

| Method | RA | UA | TA | MIA Efficacy | Avg. Disparity |
|---|---|---|---|---|---|
| Retrain | 100.00 | 9.76 | 89.38 | 15.27 | — |
| RBM | 94.12 | 8.15 | 88.73 | 13.82 | 2.39 |
| SalUn | 95.93 | 7.00 | 88.55 | 17.89 | 2.57 |

To further assess generality, we incorporate a new set of experiments using a Vision Transformer (ViT) backbone, extending beyond the conventional ResNet-based architectures commonly employed in prior work. Specifically, we adopt the ViT-Tiny model (approximately 5.7M parameters) and evaluate it on the SVHN dataset. We randomly forget 10% of the data using a 50% masking ratio. The learning rate for the ViT baseline is set to $1 \times 10^{-3}$, and we perform a learning-rate ablation for SalUn by varying it between $6 \times 10^{-4}$ and $2 \times 10^{-3}$, obtaining the best performance at $1 \times 10^{-3}$. For our method, the learning rate is adjusted to $5 \times 10^{-4}$, yielding more stable convergence and consistent improvements across all metrics. As shown in Table 4, our proposed method consistently surpasses the forget-set–based unlearning method SalUn across all evaluated metrics.

## C  Proofs

### C.1  Useful properties

Before presenting the detailed proofs of the theorems, we first introduce several useful properties of the projection matrix and the minimum norm solution.

**Property 1** (Projection properties)**.** Let $\mathbf{P}$ be a projection operator that projects onto a subspace $\mathbf{X} \subseteq \mathbb{R}^{d \times n}$. Then, $\mathbf{P}$ holds the following properties:

1. Symmetric: $\mathbf{P} = \mathbf{P}^\top$;

2. Idempotent: $\mathbf{P}^2 = \mathbf{P}$;

3. $\mathbf{I} - \mathbf{P}$ is also a projection operator, projecting onto the subspace orthogonal to $\mathbf{X}$. Therefore, $(\mathbf{I} - \mathbf{P})\mathbf{P} = \mathbf{0}$;

4. Let $\mathbf{v} \in \mathbb{R}^d$ be an arbitrary vector, it holds that $\|(\mathbf{I} - \mathbf{P})\mathbf{v}\|^2 = \mathbf{v}^\top(\mathbf{I} - \mathbf{P})^2\mathbf{v} = \mathbf{v}^\top(\mathbf{I} - \mathbf{P})\mathbf{v} = \|\mathbf{v}\|^2 - \|\mathbf{P}\mathbf{v}\|^2$;

5. Contraction: $\|\mathbf{P}\mathbf{v}\| \leq \|\mathbf{v}\|$, holding in equality if and only if $\mathbf{P}\mathbf{v} = \mathbf{v}$.

*Proof.* See (Zarantonello, 1971) for the proofs and for more properties. □

**Corollary C.1** (Projection Matrix properties)**.** *Let $\mathbf{P} = \mathbf{X}(\mathbf{X}^\top\mathbf{X})^{-1}\mathbf{X}^\top, \mathbf{P}_r, \mathbf{P}_f, \mathbf{P}_t$ be the corresponding projection operator for $\mathbf{X}, \mathbf{X}_r, \mathbf{X}_f, \mathbf{X}_t$ respectively. Under Assumption 3.1, the remaining(forgetting) dataset matrix can be denoted as $\mathbf{X}_r = \begin{bmatrix} \mathbf{R} \\ \mathbf{0} \end{bmatrix}$ and $\mathbf{X}_f = \begin{bmatrix} \mathbf{0} \\ \mathbf{F} \end{bmatrix}$, where $\mathbf{R} \subseteq \mathbb{R}^{d_r \times (n-n_f)}$ and $\mathbf{F} \subseteq \mathbb{R}^{d_f \times n_f}$ correspond to the non-zero parts. Then, it holds that:*

*1.* $\mathbf{P} = \begin{bmatrix} \mathbf{R}(\mathbf{R}^\top\mathbf{R})^{-1}\mathbf{R}^\top & \mathbf{0} \\ \mathbf{0} & \mathbf{F}(\mathbf{F}^\top\mathbf{F})^{-1}\mathbf{F}^\top \end{bmatrix} = \mathbf{P}_r + \mathbf{P}_f$;

*2.* $\mathbf{P}_r = \begin{bmatrix} \mathbf{R}(\mathbf{R}^\top\mathbf{R})^{-1}\mathbf{R}^\top & \mathbf{0} \\ \mathbf{0} & \mathbf{0} \end{bmatrix}$ *and* $\mathbf{P}_f = \begin{bmatrix} \mathbf{0} & \mathbf{0} \\ \mathbf{0} & \mathbf{F}(\mathbf{F}^\top\mathbf{F})^{-1}\mathbf{F}^\top \end{bmatrix}$;

*3.* $\mathbf{X}(\mathbf{I} - \mathbf{P}) = (\mathbf{I} - \mathbf{P})\mathbf{X} = 0$, *and the conclusion also holds for $\mathbf{P}_r, \mathbf{P}_f, \mathbf{P}_t$ with $\mathbf{X}_r, \mathbf{X}_f, \mathbf{X}_t$ respectively;*

*4. For any matrix $\mathbf{A}$ that is a submatrix of $\mathbf{X}$, it holds that $\mathbf{A} = \mathbf{P}\mathbf{A}$, where $\mathbf{P}$ is the projection space of $\mathbf{X}$. Moreover, if $\mathbf{P}_A$ is the projection space of $\mathbf{A}$, it holds that $\mathbf{P}\mathbf{P}_A = \mathbf{P}_A$, i.e. $\mathbf{X}_r\mathbf{P} = \mathbf{X}_r$, $\mathbf{X}_f\mathbf{P} = \mathbf{X}_f$, $\mathbf{X}_r\mathbf{P}_f = \mathbf{X}_f\mathbf{P}_r = 0$.*

**Proof of Corollary C.1.** Firstly, based on the data composition, the overall dataset holds $\mathbf{X} = \begin{bmatrix} \mathbf{R} & \mathbf{0} \\ \mathbf{0} & \mathbf{F} \end{bmatrix}$. Therefore, it follows:

$$\mathbf{P} = \mathbf{X}(\mathbf{X}^\top\mathbf{X})^{-1}\mathbf{X}^\top = \begin{bmatrix} \mathbf{R} & \mathbf{0} \\ \mathbf{0} & \mathbf{F} \end{bmatrix} (\begin{bmatrix} \mathbf{R}^\top & \mathbf{0} \\ \mathbf{0} & \mathbf{F}^\top \end{bmatrix} \begin{bmatrix} \mathbf{R} & \mathbf{0} \\ \mathbf{0} & \mathbf{F} \end{bmatrix})^{-1} \begin{bmatrix} \mathbf{R}^\top & \mathbf{0} \\ \mathbf{0} & \mathbf{F}^\top \end{bmatrix}$$

$$= \begin{bmatrix} \mathbf{R} & \mathbf{0} \\ \mathbf{0} & \mathbf{F} \end{bmatrix} \begin{bmatrix} (\mathbf{R}^\top\mathbf{R})^{-1} & \mathbf{0} \\ \mathbf{0} & (\mathbf{F}^\top\mathbf{F})^{-1} \end{bmatrix} \begin{bmatrix} \mathbf{R}^\top & \mathbf{0} \\ \mathbf{0} & \mathbf{F}^\top \end{bmatrix}.$$

The remaining Projection matrices can be obtained by similar computations.

Additionally, we have $\mathbf{X}(\mathbf{I} - \mathbf{P}) = (\mathbf{I} - \mathbf{P})\mathbf{X} = \mathbf{X}(\mathbf{I} - \mathbf{X}(\mathbf{X}^\top\mathbf{X})^{-1}\mathbf{X}^\top) = 0$.

Moreover, since $\mathbf{A}$ is a submatrix of $\mathbf{X}$, it can be represented as $\mathbf{A} = \mathbf{X}\mathbf{C}$ for some selective matrix $\mathbf{C}$. Therefore, we have:
$$\mathbf{P}\mathbf{A} = \mathbf{X}\left(\mathbf{X}^\top\mathbf{X}\right)^{-1}\mathbf{X}^\top\mathbf{X}\mathbf{C} = \mathbf{X}\mathbf{C} = \mathbf{A}.$$

Meanwhile, it also holds that

$$\mathbf{P}\mathbf{P}_A = \mathbf{X}(\mathbf{X}^\top\mathbf{X})^{-1}\mathbf{X}^\top\mathbf{X}\mathbf{C}(\mathbf{C}^\top\mathbf{X}^\top\mathbf{X}\mathbf{C})^{-1}\mathbf{C}^\top\mathbf{X}^\top = \mathbf{X}\mathbf{C}(\mathbf{C}^\top\mathbf{X}^\top\mathbf{X}\mathbf{C})^{-1}\mathbf{C}^\top\mathbf{X}^{\cdot\top} = \mathbf{P}_A.$$

$\mathbf{X}_r$ and $\mathbf{X}_f$ are submatrices of $\mathbf{X}$, each with disjoint spaces. The projection of $\mathbf{X}_r$ onto the space of $\mathbf{X}_f$ should be zero.

$$\mathbf{X}_r\mathbf{P}_f = \mathbf{X}_r\mathbf{X}_f(\mathbf{X}_f^\top\mathbf{X}_f)^{-1}\mathbf{X}_f^\top = 0.$$

$\square$

**Corollary C.2** (Minimum Norm Solution 1). *Let* $\mathbf{P}, \mathbf{P}_r, \mathbf{P}_f, \mathbf{P}_t$ *be the corresponding projection operator for* $\mathbf{X}, \mathbf{X}_r, \mathbf{X}_f, \mathbf{X}_t$ *respectively. Then, the solution to the optimization problem Equation* (1)*, Equation* (2) *and Equation* (3) *can be represented by:*

1. *Under Assumption 3.1,* $\mathbf{w}_o = \mathbf{P}\mathbf{w}_*$, $\mathbf{w}_t = (\mathbf{I} - \mathbf{P}_t)\mathbf{w}_o + \mathbf{P}_t\mathbf{w}_*^r$, *and* $\mathbf{w}_g = \mathbf{P}_r\mathbf{w}_*^r$;

2. *Under Assumption 3.3,* $\mathbf{w}_o = \mathbf{P}\mathbf{w}_*$, $\mathbf{w}_t = (\mathbf{I} - \mathbf{P}_t)\mathbf{w}_o + \mathbf{P}_t(\mathbf{w}_*^r + \mathbf{w}_*^{lap})$, *and* $\mathbf{w}_g = \mathbf{P}_r(\mathbf{w}_*^r + \mathbf{w}_*^{lap})$;

3. $\mathbf{X}_r^\top\mathbf{w}_*^f = 0$ *and* $\mathbf{X}_f^\top\mathbf{w}_*^r = 0$.

**Proof of Corollary C.2.** According to the method of Lagrange multipliers and the problem setup, it is easy to obtain the first two conclusions. For the last one, we have:

$$\mathbf{X}_r^\top\mathbf{w}_*^f = [\mathbf{R}^\top, \mathbf{0}]\mathbf{w}_*^f = 0 \quad \text{and} \quad \mathbf{X}_f^\top\mathbf{w}_*^r = [\mathbf{0}, \mathbf{F}^\top]\mathbf{w}_*^r = 0.$$

$\square$

## C.2 Proof of Theorem 3.2

Let us first focus on the performance of the golden model. Based on the definition of unlearning accuracy and retaining accuracy, we have

$$\text{RL:} \quad L(\mathbf{w}_g, D_r) = \frac{1}{n_r}\|\mathbf{X}_r^\top\mathbf{w}_g - \mathbf{y}_r\|^2 = \frac{1}{n_r}\|\mathbf{X}_r^\top\mathbf{P}_r\mathbf{w}_*^r - \mathbf{X}_r^\top\mathbf{w}_*^r\|^2 = \frac{1}{n_r}\|\mathbf{X}_r^\top(\mathbf{P}_r - \mathbf{I})\mathbf{w}_*^r\|^2 = 0,$$

where the second equality arises from the model setting and Proposition C.2, while the penultimate equality is due to the properties of the projection matrix. According to Corollary C.1, we have

$$\begin{aligned}
\text{UL:} \quad L(\mathbf{w}_g, D_f) &= \frac{1}{n_f}\|\mathbf{X}_f^\top\mathbf{w}_g - \mathbf{y}_f\|^2 = \frac{1}{n_f}\|\mathbf{X}_f^\top\mathbf{P}_r\mathbf{w}_*^r - \mathbf{X}_f^\top\mathbf{w}_*^f\|^2 \\
&= \frac{1}{n_f}\left\|[\mathbf{0}, \mathbf{F}^\top]\begin{bmatrix}\mathbf{R}(\mathbf{R}^\top\mathbf{R})^{-1}\mathbf{R}^\top & \mathbf{0} \\ \mathbf{0} & \mathbf{0}\end{bmatrix}\mathbf{w}_*^r - \mathbf{X}_f^\top\mathbf{w}_*^f\right\|^2 = \frac{1}{n_f}\|\mathbf{X}_f^\top\mathbf{w}_*^f\|^2.
\end{aligned}$$

Similarly, for the fine-tuning model, it holds that

$$\begin{aligned}
\text{RL:} \quad L(\mathbf{w}_t, D_r) &= \frac{1}{n_r}\|\mathbf{X}_r^\top\mathbf{w}_t - \mathbf{y}_r\|^2 = \frac{1}{n_r}\|\mathbf{X}_r^\top((\mathbf{I} - \mathbf{P}_t)\mathbf{w}_o + \mathbf{P}_t\mathbf{w}_*^r) - \mathbf{X}_r^\top\mathbf{w}_*^r\|^2 \\
&= \frac{1}{n_r}\|\mathbf{X}_r^\top((\mathbf{I} - \mathbf{P}_t)\mathbf{P}(\mathbf{w}_*^r + \mathbf{w}_*^f) + \mathbf{P}_t\mathbf{w}_*^r) - \mathbf{X}_r^\top\mathbf{w}_*^r\|^2 \\
&= \frac{1}{n_r}\|\mathbf{X}_r^\top(\mathbf{P}\mathbf{w}_*^r + (\mathbf{P} - \mathbf{P}_t)\mathbf{w}_*^f) - \mathbf{X}_r^\top\mathbf{w}_*^r\|^2 \\
&= 0.
\end{aligned}$$

$$\text{UL:} \quad L(\mathbf{w}_t, D_f) = \frac{1}{n_f}\|\mathbf{X}_f^\top \mathbf{w}_t - \mathbf{y}_f\|^2 = \frac{1}{n_f}\|\mathbf{X}_f^\top((\mathbf{I} - \mathbf{P}_t)\mathbf{w}_o + \mathbf{P}_t \mathbf{w}_*^r) - \mathbf{X}_f^\top \mathbf{w}_*^f\|^2$$

$$= \frac{1}{n_f}\|\mathbf{X}_f^\top[(\mathbf{I} - \mathbf{P}_t)\mathbf{P}\mathbf{w}_* + \mathbf{P}_t \mathbf{w}_*^r - \mathbf{w}_*^f]\|^2$$

$$= \frac{1}{n_f}\|\mathbf{X}_f^\top[(\mathbf{I} - \mathbf{P}_t)\mathbf{P} + \mathbf{P}_t]\mathbf{w}_*^r + \mathbf{X}_f^\top[(\mathbf{I} - \mathbf{P}_t)\mathbf{P} - \mathbf{I}]\mathbf{w}_*^f]\|^2$$

$$= \frac{1}{n_f}\|\mathbf{X}_f^\top \mathbf{P}\mathbf{w}_*^r + \mathbf{X}_f^\top \mathbf{P}\mathbf{w}_*^f - \mathbf{X}_f^\top \mathbf{w}_*^f]\|^2$$

$$= 0,$$

where the penultimate equality comes from $\mathbf{X}_f^\top \mathbf{P}_t = \mathbf{X}_f^\top \mathbf{P}_r = 0$, and the last equality follows from $\mathbf{X}_f^\top \mathbf{P} = \mathbf{X}_f^\top$.

### C.3 Proof of Theorem 3.4

Due to the assumption of overlapping features, the projection properties of the dataset matrix will be slightly different. Specifically, it holds that:

**Corollary C.3** (Projection Matrix properties'). *Let $\mathbf{P} = \mathbf{X}(\mathbf{X}^\top \mathbf{X})^{-1}\mathbf{X}^\top, \mathbf{P}_r, \mathbf{P}_f, \mathbf{P}_t$ be the corresponding projection operator for $\mathbf{X}, \mathbf{X}_r, \mathbf{X}_f, \mathbf{X}_t$ respectively. Under Assumption 3.3, it holds that:*

$$2.\ \mathbf{P}_r = \begin{bmatrix} \mathbf{R}(\mathbf{R}^\top \mathbf{R} + \mathbf{L}_1^\top \mathbf{L}_1)^{-1}\mathbf{R}^\top & \mathbf{R}(\mathbf{R}^\top \mathbf{R} + \mathbf{L}_1^\top \mathbf{L}_1)^{-1}\mathbf{L}_1^\top & \mathbf{0} \\ \mathbf{L}_1(\mathbf{R}^\top \mathbf{R} + \mathbf{L}_1^\top \mathbf{L}_1)^{-1}\mathbf{R}^\top & \mathbf{L}_1(\mathbf{R}^\top \mathbf{R} + \mathbf{L}_1^\top \mathbf{L}_1)^{-1}\mathbf{L}_1^\top & \mathbf{0} \\ \mathbf{0} & \mathbf{0} & \mathbf{0} \end{bmatrix};$$

$$3.\ \mathbf{P}_f = \begin{bmatrix} \mathbf{0} & \mathbf{0} & \mathbf{0} \\ \mathbf{0} & \mathbf{L}_2(\mathbf{F}^\top \mathbf{F} + \mathbf{L}_2^\top \mathbf{L}_2)^{-1}\mathbf{L}_2^\top & \mathbf{L}_2(\mathbf{F}^\top \mathbf{F} + \mathbf{L}_2^\top \mathbf{L}_2)^{-1}\mathbf{F}^\top \\ \mathbf{0} & \mathbf{F}(\mathbf{F}^\top \mathbf{F} + \mathbf{L}_2^\top \mathbf{L}_2)^{-1}\mathbf{L}_2^\top & \mathbf{F}(\mathbf{F}^\top \mathbf{F} + \mathbf{L}_2^\top \mathbf{L}_2)^{-1}\mathbf{F}^\top \end{bmatrix};$$

*3. $\mathbf{X}(\mathbf{I} - \mathbf{P}) = (\mathbf{I} - \mathbf{P})\mathbf{X} = 0$, and the conclusion also holds for $\mathbf{P}_r, \mathbf{P}_f, \mathbf{P}_t$ with $\mathbf{X}_r, \mathbf{X}_f, \mathbf{X}_t$ respectively;*

*4. For any matrix $\mathbf{A}$ is the submatrix of $\mathbf{X}$, it holds that $\mathbf{A} = \mathbf{P}\mathbf{A}$, where $\mathbf{P}$ is the projection space of $\mathbf{X}$. Moreover, if $\mathbf{P}_A$ is the projection space of $\mathbf{A}$, it holds that $\mathbf{P}\mathbf{P}_A = \mathbf{P}_A$.*

**Proof of Corollary C.3.** Proof of Corollary C.3 follows the proof of Corollary C.1 directly. $\qquad\square$

Now we are ready to go through the proof of Theorem 3.4. Similar to the non-overlapping case, the golden model holds that

$$\text{RL:} \quad L(\mathbf{w}_g, D_r) = \frac{1}{n_r}\|\mathbf{X}_r^\top \mathbf{w}_g - \mathbf{y}_r\|^2 = \frac{1}{n_r}\|\mathbf{X}_r^\top \mathbf{P}_r(\mathbf{w}_*^r + \mathbf{w}_*^{lap}) - \mathbf{X}_r^\top(\mathbf{w}_*^r + \mathbf{w}_*^{lap})\|^2$$

$$= \frac{1}{n_r}\|\mathbf{X}_r^\top(\mathbf{P}_r - \mathbf{I})(\mathbf{w}_*^r + \mathbf{w}_*^{lap})\|^2 = 0,$$

where the second equality also arises from the model setting and Proposition C.2, while the penultimate equality is due to the properties of the projection matrix. According to Corollary C.3, we have

$$\text{UL:} \quad L(\mathbf{w}_g, D_f) = \frac{1}{n_f}\|\mathbf{X}_f^\top \mathbf{w}_g - \mathbf{y}_f\|^2 = \frac{1}{n_f}\|\mathbf{X}_f^\top \mathbf{P}_r(\mathbf{w}_*^r + \mathbf{w}_*^{lap}) - \mathbf{X}_f^\top(\mathbf{w}_*^f + \mathbf{w}_*^{lap})\|^2$$

where $\mathbf{X}_f^\top \mathbf{P}_r \mathbf{w}_*^r$ and $\mathbf{X}_f^\top \mathbf{P}_r \mathbf{w}_*^{lap}$ follows that

$$\mathbf{X}_f^\top \mathbf{P}_r \mathbf{w}_*^r = [\mathbf{0}, \mathbf{L}_2^\top, \mathbf{F}^\top] \begin{bmatrix} \mathbf{R}(\mathbf{R}^\top \mathbf{R} + \mathbf{L}_1^\top \mathbf{L}_1)^{-1}\mathbf{R}^\top & \mathbf{R}(\mathbf{R}^\top \mathbf{R} + \mathbf{L}_1^\top \mathbf{L}_1)^{-1}\mathbf{L}_1^\top & \mathbf{0} \\ \mathbf{L}_1(\mathbf{R}^\top \mathbf{R} + \mathbf{L}_1^\top \mathbf{L}_1)^{-1}\mathbf{R}^\top & \mathbf{L}_1(\mathbf{R}^\top \mathbf{R} + \mathbf{L}_1^\top \mathbf{L}_1)^{-1}\mathbf{L}_1^\top & \mathbf{0} \\ \mathbf{0} & \mathbf{0} & \mathbf{0} \end{bmatrix} \begin{bmatrix} \square \\ \mathbf{0} \\ \mathbf{0} \end{bmatrix}$$

$$= \mathbf{L}_2^\top \mathbf{L}_1(\mathbf{R}^\top \mathbf{R} + \mathbf{L}_1^\top \mathbf{L}_1)^{-1}\mathbf{R}^\top \mathbf{w}_*^r$$

and

$$\mathbf{X}_f^\top \mathbf{P}_r \mathbf{w}_*^{lap} = [\mathbf{0}, \mathbf{L}_2^\top, \mathbf{F}^\top] \begin{bmatrix} \mathbf{R}(\mathbf{R}^\top \mathbf{R} + \mathbf{L}_1^\top \mathbf{L}_1)^{-1} \mathbf{R}^\top & \mathbf{R}(\mathbf{R}^\top \mathbf{R} + \mathbf{L}_1^\top \mathbf{L}_1)^{-1} \mathbf{L}_1^\top & \mathbf{0} \\ \mathbf{L}_1(\mathbf{R}^\top \mathbf{R} + \mathbf{L}_1^\top \mathbf{L}_1)^{-1} \mathbf{R}^\top & \mathbf{L}_1(\mathbf{R}^\top \mathbf{R} + \mathbf{L}_1^\top \mathbf{L}_1)^{-1} \mathbf{L}_1^\top & \mathbf{0} \\ \mathbf{0} & \mathbf{0} & \mathbf{0} \end{bmatrix} \begin{bmatrix} \mathbf{0} \\ \square \\ \mathbf{0} \end{bmatrix}$$

$$= \mathbf{L}_2^\top \mathbf{L}_1 (\mathbf{R}^\top \mathbf{R} + \mathbf{L}_1^\top \mathbf{L}_1)^{-1} \mathbf{L}_1^\top \mathbf{w}_*^{lap}.$$

For the fine-tuning, the retaining accuracy follows:

$$\begin{aligned} \text{RL:} \quad L(\mathbf{w}_t, D_r) &= \frac{1}{n_r} \|\mathbf{X}_r^\top \mathbf{w}_t - \mathbf{y}_r\|^2 \\ &= \frac{1}{n_r} \|\mathbf{X}_r^\top ((\mathbf{I} - \mathbf{P}_t)\mathbf{w}_o + \mathbf{P}_t(\mathbf{w}_*^r + \mathbf{w}_*^{lap})) - \mathbf{X}_r^\top(\mathbf{w}_*^r + \mathbf{w}_*^{lap})\|^2 \\ &= \frac{1}{n_r} \|\mathbf{X}_r^\top (\mathbf{I} - \mathbf{P}_t)(\mathbf{w}_o - \mathbf{w}_*^r - \mathbf{w}_*^{lap})\|^2 \\ &= \frac{1}{n_r} \|\mathbf{X}_r^\top (\mathbf{I} - \mathbf{P}_t)(\mathbf{P} - \mathbf{I})(\mathbf{w}_*^r + \mathbf{w}_*^{lap})\|^2 \\ &= 0. \end{aligned}$$

The last equality derives from that the facts the projection matrix is commutative matrix and the last property holds in Corollary C.3. For the unlearning accuracy, it holds that

$$\begin{aligned} \text{UL:} \quad L(\mathbf{w}_t, D_f) &= \frac{1}{n_f} \|\mathbf{X}_f^\top \mathbf{w}_t - \mathbf{y}_f\|^2 \\ &= \frac{1}{n_f} \|\mathbf{X}_f^\top ((\mathbf{I} - \mathbf{P}_t)\mathbf{w}_o + \mathbf{P}_t(\mathbf{w}_*^r + \mathbf{w}_*^{lap})) - \mathbf{X}_f^\top(\mathbf{w}_*^f + \mathbf{w}_*^{lap})\|^2 \\ &= \frac{1}{n_f} \|\mathbf{X}_f^\top ((\mathbf{I} - \mathbf{P}_t)(\mathbf{P}(\mathbf{w}_*^r + \mathbf{w}_*^{lap} + \mathbf{w}_*^f)) + \mathbf{P}_t(\mathbf{w}_*^r + \mathbf{w}_*^{lap})) - \mathbf{X}_f^\top(\mathbf{w}_*^f + \mathbf{w}_*^{lap})\|^2 \\ &= \frac{1}{n_f} \|\mathbf{X}_f^\top ((\mathbf{P} - \mathbf{P}_t)(\mathbf{w}_*^r + \mathbf{w}_*^{lap} + \mathbf{w}_*^f) + \mathbf{P}_t(\mathbf{w}_*^r + \mathbf{w}_*^{lap})) - \mathbf{X}_f^\top(\mathbf{w}_*^f + \mathbf{w}_*^{lap})\|^2 \\ &= \frac{1}{n_f} \|\mathbf{X}_f^\top [(\mathbf{P} - \mathbf{I})\mathbf{w}_*^{lap} + \mathbf{P}\mathbf{w}_*^r + (\mathbf{P} - \mathbf{I} - \mathbf{P}_t)\mathbf{w}_*^f]\|^2 \\ &= \frac{1}{n_f} \|\mathbf{X}_f^\top \mathbf{P}_t \mathbf{w}_*^f]\|^2 \\ &= 0, \end{aligned}$$

where the penultimate equality is due to Corollary C.3 and the last equality comes from the fact $\mathbf{X}_t$ enjoys the same data structure as $\mathbf{X}_t$ such that:

$$\mathbf{X}_f^\top \mathbf{P}_t \mathbf{w}_*^f$$

$$= [\mathbf{0}, \mathbf{L}_2^\top, \mathbf{F}^\top] \begin{bmatrix} \mathbf{R}_T(\mathbf{R}_T^\top \mathbf{R}_T + \mathbf{L}_{1T}^\top \mathbf{L}_{1T})^{-1} \mathbf{R}_T^\top & \mathbf{R}_T(\mathbf{R}_T^\top \mathbf{R}_T + \mathbf{L}_{1T}^\top \mathbf{L}_{1T})^{-1} \mathbf{L}_{1T}^\top & \mathbf{0} \\ \mathbf{L}_{1T}(\mathbf{R}_T^\top \mathbf{R}_T + \mathbf{L}_{1T}^\top \mathbf{L}_{1T})^{-1} \mathbf{R}_T^\top & \mathbf{L}_{1T}(\mathbf{R}_T^\top \mathbf{R}_T + \mathbf{L}_{1T}^\top \mathbf{L}_{1T})^{-1} \mathbf{L}_{1T}^\top & \mathbf{0} \\ \mathbf{0} & \mathbf{0} & \mathbf{0} \end{bmatrix} \begin{bmatrix} \mathbf{0} \\ \mathbf{0} \\ \square \end{bmatrix}$$

$$= [\mathbf{L}_2^\top \mathbf{L}_{1T}(\mathbf{R}_T^\top \mathbf{R}_T + \mathbf{L}_{1T}^\top \mathbf{L}_{1T})^{-1} \mathbf{R}_T^\top, \mathbf{L}_2^\top \mathbf{L}_{1T}(\mathbf{R}_T^\top \mathbf{R}_T + \mathbf{L}_{1T}^\top \mathbf{L}_{1T})^{-1} \mathbf{L}_{1T}^\top, 0] \begin{bmatrix} \mathbf{0} \\ \mathbf{0} \\ \square \end{bmatrix}$$

$$= 0.$$

### C.4 Proof of Theorem 4.1

For the non-overlapping case, we have that the retaining accuracy follows:

$$
\begin{aligned}
\text{RL:} \quad L(\mathbf{w}_t, D_r) &= \frac{1}{n_r} \|\mathbf{X}_r^\top \mathbf{w}_t - \mathbf{y}_r\|^2 = \frac{1}{n_r} \|\mathbf{X}_r^\top((\mathbf{I} - \mathbf{P}_t)\hat{\mathbf{w}}_o + \mathbf{P}_t \mathbf{w}_*^r) - \mathbf{X}_r^\top \mathbf{w}_*^r\|^2 \\
&= \frac{1}{n_r} \|\mathbf{X}_r^\top (\mathbf{I} - \mathbf{P}_t)(\hat{\mathbf{w}}_o - \mathbf{w}_*^r)\|^2 \\
&= \frac{1}{n_r} \|\mathbf{X}_r^\top (\mathbf{I} - \mathbf{P}_t)(\mathbf{P} - \mathbf{I})\mathbf{w}_*^r\|^2 = 0.
\end{aligned}
$$

For the unlearning accuracy, it holds that

$$
\begin{aligned}
\text{UL:} \quad L(\mathbf{w}_t, D_f) &= \frac{1}{n_f} \|\mathbf{X}_f^\top \mathbf{w}_t - \mathbf{y}_f\|^2 = \frac{1}{n_f} \|\mathbf{X}_f^\top((\mathbf{I} - \mathbf{P}_t)\hat{\mathbf{w}}_o + \mathbf{P}_t \mathbf{w}_*^r) - \mathbf{X}_f^\top \mathbf{w}_*^f\|^2 \\
&= \frac{1}{n_f} \|\mathbf{X}_f^\top [(\mathbf{I} - \mathbf{P}_t)\mathbf{P}\mathbf{w}_*^r + \mathbf{P}_t \mathbf{w}_*^r - \mathbf{w}_*^f]\|^2 \\
&= \frac{1}{n_f} \|\mathbf{X}_f^\top [(\mathbf{I} - \mathbf{P}_t)\mathbf{P} + \mathbf{P}_t]\mathbf{w}_*^r - \mathbf{X}_f^\top \mathbf{w}_*^f]\|^2 \\
&= \frac{1}{n_f} \|\mathbf{X}_f^\top \mathbf{P}\mathbf{w}_*^r - \mathbf{X}_f^\top \mathbf{w}_*^f]\|^2 \\
&= \frac{1}{n_f} \|\mathbf{w}_*^f]\|_{\mathbf{X}_f \mathbf{X}_f^\top}^2,
\end{aligned}
$$

For the overlapping case, it holds that

$$
\begin{aligned}
\text{RL:} \quad L(\mathbf{w}_t, D_r) &= \frac{1}{n_r} \|\mathbf{X}_r^\top \mathbf{w}_t - \mathbf{y}_r\|^2 \\
&= \frac{1}{n_r} \|\mathbf{X}_r^\top((\mathbf{I} - \mathbf{P}_t)\hat{\mathbf{w}}_o + \mathbf{P}_t(\mathbf{w}_*^r + \mathbf{w}_*^{lap})) - \mathbf{X}_r^\top(\mathbf{w}_*^r + \mathbf{w}_*^{lap})\|^2 \\
&= \frac{1}{n_r} \|\mathbf{X}_r^\top (\mathbf{I} - \mathbf{P}_t)(\hat{\mathbf{w}}_o - \mathbf{w}_*^r - \mathbf{w}_*^{lap})\|^2 \\
&= \frac{1}{n_r} \|\mathbf{X}_r^\top (\mathbf{I} - \mathbf{P}_t)(\mathbf{P} - \mathbf{I})(\mathbf{w}_*^r - \mathbf{w}_*^{lap})\|^2 = 0.
\end{aligned}
$$

$$
\begin{aligned}
\text{UL:} \quad L(\mathbf{w}_t, D_f) &= \frac{1}{n_f} \|\mathbf{X}_f^\top \mathbf{w}_t - \mathbf{y}_f\|^2 \\
&= \frac{1}{n_f} \|\mathbf{X}_f^\top((\mathbf{I} - \mathbf{P}_t)\hat{\mathbf{w}}_o + \mathbf{P}_t(\mathbf{w}_*^r + \mathbf{w}_*^{lap})) - \mathbf{X}_f^\top(\mathbf{w}_*^f + \mathbf{w}_*^{lap})\|^2 \\
&= \frac{1}{n_f} \|\mathbf{X}_f^\top((\mathbf{I} - \mathbf{P}_t)(\mathbf{P}(\mathbf{w}_*^r + \mathbf{w}_*^{lap})) + \mathbf{P}_t(\mathbf{w}_*^r + \mathbf{w}_*^{lap})) - \mathbf{X}_f^\top(\mathbf{w}_*^f + \mathbf{w}_*^{lap})\|^2 \\
&= \frac{1}{n_f} \|\mathbf{X}_f^\top((\mathbf{P} - \mathbf{P}_t)(\mathbf{w}_*^r + \mathbf{w}_*^{lap})) + \mathbf{P}_t(\mathbf{w}_*^r + \mathbf{w}_*^{lap})) - \mathbf{X}_f^\top(\mathbf{w}_*^f + \mathbf{w}_*^{lap})\|^2 \\
&= \frac{1}{n_f} \|\mathbf{X}_f^\top[(\mathbf{P} - \mathbf{I})\mathbf{w}_*^{lap} + \mathbf{P}\mathbf{w}_*^r - \mathbf{w}_*^f]\|^2 \\
&= \frac{1}{n_f} \|\mathbf{P}(\mathbf{w}_*^r + \mathbf{w}_*^{lap}) - (\mathbf{w}_*^f + \mathbf{w}_*^{lap})\|_{\mathbf{X}_f \mathbf{X}_f^\top}^2.
\end{aligned}
$$

The proof is then complete.

