# OpenReview forum: "Understanding Fine-tuning in Approximate Unlearning: A Theoretical Perspective"
_TMLR — Accepted by TMLR_

### Review · Reviewer_GWBK · 2025-09-10

**Summary Of Contributions:**

This paper presents a theoretical analysis of fine-tuning methods for approximate machine unlearning, focusing on the overparameterized linear regression setting. The authors investigate why naive fine-tuning fails to effectively unlearn targeted data and propose a novel masking strategy called Retention-Based Masking (RBM) to address this issue. The paper provides theoretical and empirical evidence showing that RBM can improve unlearning accuracy while preserving retaining accuracy, particularly in scenarios with overlapping features between the forgetting and remaining datasets.

**Audience:**

Yes

**Audience Explanation:**

This paper makes a solid contribution to the field of machine "unlearning" by providing a much-needed theoretical foundation for understanding fine-tuning-based approaches. The proposed RBM method is well-motivated and shows promising results.

**Broader Impact Concerns:**

Unlearning is often motivated by ethical concerns (copyright, bias, etc.) in the training data. Overselling the capability of models to unlearn, could lead to broader impact concerns. There is no Broader Impact Statement section in the paper.

**Claims And Evidence:**

Yes

**Claims Explanation:**

The paper provides the first theoretical analysis of fine-tuning for machine unlearning in the linear regression setting. The proposed RBM strategy is a direct and logical consequence of the theoretical analysis. The insight that masking should be based on the remaining data to preserve shared features is novel and well-supported by the theoretical results. The paper includes experiments on both synthetic and real-world datasets (CIFAR-10) to validate the theoretical claims. The synthetic experiments are particularly effective in illustrating the theoretical concepts of distinct and overlapping features.

**Requested Changes:**

The theoretical analysis is confined to the overparameterized linear regression setting. While this provides a valuable starting point, the extent to which these insights generalize to more complex, non-linear models is not immediately clear. The authors should discuss the potential challenges and future directions for extending their theory to more practical settings.

The analysis relies on strong assumptions, particularly the "distinct features" and "overlapping features". While these assumptions are useful for simplifying the analysis, they may not accurately reflect the complexities of real-world data. A discussion of the limitations of these assumptions would strengthen the paper.

The paper primarily focuses on fine-tuning and masking-based methods. While it provides a good overview of related work, a more direct comparison with other approximate unlearning techniques like gradient ascent, influence unlearning, in the experimental section would provide a more complete picture of RBM's performance.

---

> ### Author Response · Authors · 2025-10-13
> **Response**
>
> We thank Reviewer GWBK for the detailed review and constructive feedback, and hope to address your concerns as follows:
>
> **Response to assumptions and models setup:** We thank the reviewer for the insightful comment regarding the assumptions and model setup, and we would like to provide further clarification here. One potential challenge in extending our conclusions to more complex models and relaxing the current assumptions lies in the fact that our feature assumptions are grounded in a high-dimensional linear regression framework, where distinct features refer to those unique to either the forgetting or retaining dataset, and overlapping features are shared between the two.
>
> For illustration, consider two categories: bananas and cars. Bananas have a distinct feature, such as an elongated shape, while cars have a unique feature like mirrors; these are distinct features. Meanwhile, both may share overlapping features such as color—for example, both can be yellow.
>
> When extending this intuition to nonlinear models, one potential way is through a feature learning framework, where features can be identified at the patch level in high-dimensional representations. For instance, a class like "banana" can be represented as $x_{\text {banana }}=[y_1 \cdot u_{d_1}, y_1 \cdot u_o]$
> and "car" as
> $x_{\text {car }}=[y_2 \cdot u_{d_2}, y_2 \cdot u_o],$
> where $y_1, y_2$ denote the corresponding class labels, $u_{d_i}$ represent the distinct feature components, and $u_o$ corresponds to the overlapping feature component.
>
> Under this data model, the same analytical framework naturally extends to high-dimensional nonlinear networks, like a two-layer neural network, allowing our theoretical conclusions to generalize beyond the linear setting.
>
> **Response to other unlearn methods:** Thank you for the suggestion. In this paper, our goal is mainly to explain when and why fine-tuning succeeds or fails at unlearning; accordingly, we emphasize FT-centric comparison rather than broadening other approximate unlearning methods. In the following, we strengthen external validity by expanding coverage across data regimes and architectures. Concretely, we will add experiments on SVHN and tiny-ImageNet and evaluate on additional backbones (e.g., a tiny ViT) to show that our theoretical conclusions. Please check the following results:
>
> Table 1: Unlearning Results on TinyImageNet
>
> | **Method** | **RA** | **UA** | **TA** | **MIA Efficacy** | **Avg. Disparity** |
> | :--------- | :----: | :----: | :----: | :--------------: | :----------------: |
> | Retrain    |  99.98 |  39.91 |  62.21 |       65.13      |          —         |
> | RBM        |  96.89 |  47.23 |  59.91 |       69.98      |        4.39        |
> | SalUn      |  97.20 |  31.30 |  59.10 |       71.08      |        5.11        |
>
>
> Table 2: SVHN Unlearning Results using ViT-Tiny
>
> | **Method** | **RA** | **UA** | **TA** | **MIA Efficacy** | **Avg. Disparity** |
> | :--------- | :----: | :----: | :----: | :--------------: | :----------------: |
> | Retrain    | 100.00 |  9.76  |  89.38 |       15.27      |          —         |
> | RBM        |  94.12 |  8.15  |  88.73 |       13.82      |        2.39        |
> | SalUn      |  95.93 |  7.00  |  88.55 |       17.89      |        2.57        |

---

### Review · Reviewer_GuEV · 2025-09-16

**Summary Of Contributions:**

This paper investigates why fine-tuning (FT), a common method for approximate machine unlearning, often fails to forget targeted data despite maintaining performance on remaining data. Through a theoretical analysis within an overparameterized linear regression framework, the authors show that FT models preserve information from the original pretrained model, preventing effective forgetting. They extend the analysis to scenarios with both distinct and overlapping features, confirming that the forgetting failure persists in both cases. To address this, they propose Retention-Based Masking (RBM), which constructs saliency maps based on the remaining dataset instead of the forgetting dataset, thereby preserving overlapping features. Experiments on synthetic and real-world datasets validate that RBM significantly improves unlearning accuracy, retaining accuracy, and fairness compared to existing fine-tuning and masking approaches.

**Audience:**

Yes

**Audience Explanation:**

Yes, the findings of this paper would be of strong interest to TMLR’s audience, as machine unlearning is a rapidly growing area motivated by privacy regulations such as GDPR and the “right to be forgotten.” The paper addresses a fundamental question about the limitations of fine-tuning for unlearning, providing both theoretical clarity and a practical improvement through Retention-Based Masking. These insights are valuable not only for researchers in privacy-preserving machine learning but also for those working on reliable and efficient model update strategies.

**Broader Impact Concerns:**

I do not see any ethical concerns in this work that would necessitate a Broader Impact Statement. The paper’s contributions focus on theoretical analysis and methodological improvements for machine unlearning, without raising issues related to fairness, misuse, or sensitive applications. As such, the ethical implications appear minimal and sufficiently addressed by the scope of the research.

**Claims And Evidence:**

Yes

**Claims Explanation:**

The paper provides a solid theoretical framework using overparameterized linear regression to explain why naive fine-tuning fails in approximate unlearning, and the proofs are carefully aligned with both distinct and overlapping feature settings. The proposed Retention-Based Masking (RBM) is supported by both mathematical analysis and empirical validation on synthetic and real-world datasets, with results showing clear improvements over existing methods. Overall, the evidence is convincing and clearly presented, though additional experiments on larger-scale or more diverse benchmarks could further strengthen the generality of the claim.

**Requested Changes:**

1. The adopted baselines are not very recent. Including baselines from the latest 2024–2025 works would better position RBM within the evolving landscape.

2. The current experiments (mainly CIFAR datasets) are somewhat limited; adding results on larger or more diverse datasets would strengthen the claim that RBM generalizes beyond small-scale benchmarks.

3. The experiments are mostly conducted using ResNet backbone, it would be interesting to see the results on other backbone, e.g., ViT.

4. Analyzing how RBM’s performance varies with different mask thresholds and degrees of feature overlap would provide stronger evidence of its robustness.

5. It would be better to give more details regarding the random forgetting and class wise forgetting scenarios.

---

> ### Author Response · Authors · 2025-10-13
> **Response**
>
> We thank Reviewer GuEV for the detailed review and constructive feedback, and hope to address your concerns as follows:
>
> **Response to recent SOTA baselines:** Thank you for the suggestion. In this paper, our goal is mainly to explain when and why fine-tuning succeeds or fails at unlearning; accordingly, we emphasize FT-centric comparison rather than broadening SOTA baselines. In the following, we strengthen external validity by expanding coverage across data regimes and architectures. Concretely, we will add experiments on SVHN and tiny-ImageNet (in addition to CIFAR-10/100) and evaluate on additional backbones (e.g., a tiny ViT) to show our theoretical conclusions, as you suggested.
>
> **Response to experiments on more datasets:**  In our address for the reviewer's concern over the limited scope of evaluation in CIFAR benchmarks, we provide the following evaluation over the TinyImagenet dataset. The dataset consists of a sub-set of the ImageNet dataset with 200 classes and 500 samples per class for training with 50 each for validation and test sets. We consider the random forgetting scenario of $10\%$ using a masking of $50\%$ for both methods. The larger space of classes and a higher visual diversity entails a demanding task. Nonetheless, our approach significantly overcomes the baseline method, albeit with a moderate decline in the test accuracy, a testament to the dataset's complexity. RBM preserves high retain set accuracy whilst effectively mitigating MIA attacks to the standards of a retrained model, hence attaining a lower average discrepancy against this standard.
>
> Table 1: Unlearning Results on TinyImageNet
> | **Method** | **RA** | **UA** | **TA** | **MIA Efficacy** | **Avg. Disparity** |
> | :--------- | :----: | :----: | :----: | :--------------: | :----------------: |
> | Retrain    |  99.98 |  39.91 |  62.21 |       65.13      |          —         |
> | RBM        |  96.89 |  47.23 |  59.91 |       69.98      |        4.39        |
> | SalUn      |  97.20 |  31.30 |  59.10 |       71.08      |        5.11        |
>
>
> **Response to experiments on more backbones and datasets:** We appreciate the reviewer’s observation. In the following we incorporate a new set of experiments using a Vision Transformer (ViT) backbone, moving beyond the conventional ResNet-based architectures commonly used in prior work. We adopt the ViT-Tiny model (approximately 5.7M parameters) and evaluate it on the SVHN dataset. The setup was to randomly forget $10\%$ of the data using a $50\%$ masking ratio. The learning rate was set to $1e-3$ for the ViT baseline, and we conduct a learning rate ablation for SalUn, varying it between $6e-4$ and $2e-3$, with the best performance at $1e-3$. For our method, the learning rate was adjusted to $5e-4$, which yielded more stable convergence and consistent improvements across metrics. As we note in following Table, our proposed method consistently overcomes the performance of a forget set based unlearning method SalUn across the metrics covered.
>
>
> Table 2: SVHN Unlearning Results using ViT-Tiny
> | **Method** | **RA** | **UA** | **TA** | **MIA Efficacy** | **Avg. Disparity** |
> | :--------- | :----: | :----: | :----: | :--------------: | :----------------: |
> | Retrain    | 100.00 |  9.76  |  89.38 |       15.27      |          —         |
> | RBM        |  94.12 |  8.15  |  88.73 |       13.82      |        2.39        |
> | SalUn      |  95.93 |  7.00  |  88.55 |       17.89      |        2.57        |

---

> > ### Author Response · Authors · 2025-10-13
> > **Continued Response**
> >
> > **Response to experiments on threshold:**
> > Thank you for the suggestion. In the following, we include an ablation over the mask threshold in Eq.(6).
> > The results in the following Table highlight how the proposed RBM method consistently outperforms SalUn over a range of values for $\gamma$ (masking ratio). The experiment was conducted over the $10\%$ forgetting scenario using the Cifar-10 dataset. The comparative performance benefits are signified by the average disparity metric. Additionally, lower unlearning accuracy (UA) indicates stronger forgetting of the targeted data, while maintaining higher retaining accuracy (RA) and test accuracy (TA), demonstrating its ability to preserve general utility with a varying level of masking. RBM exhibits a lower average disparity, indicating a balanced unlearning–retention trade-off.
> >
> > Table 3: Gamma Ablations on CIFAR-10
> > |  **γ**  | **Method**  | **UA** | **MIA-Efficacy** | **RA** | **TA** | **Avg. Disparity** |
> > | :-----: | :---------- | :----: | :--------------: | :----: | :----: | :----------------: |
> > |    —    | **Retrain** |  5.80  |       13.91      | 100.00 |  94.30 |        0.00        |
> > | **0.2** | RBM         |  3.05  |       13.47      |  99.84 |  94.26 |        0.85        |
> > |         | SalUn       |  2.40  |       14.42      |  99.74 |  94.36 |        1.06        |
> > | **0.4** | RBM         |  3.00  |       14.07      |  99.78 |  94.02 |        0.86        |
> > |         | SalUn       |  3.84  |       14.24      |  99.01 |  93.27 |        1.08        |
> > | **0.6** | RBM         |  3.09  |       13.71      |  99.62 |  93.65 |        0.99        |
> > |         | SalUn       |  4.47  |       13.84      |  98.29 |  92.67 |        1.19        |
> > | **0.7** | RBM         |  3.18  |       13.16      |  99.51 |  93.91 |        1.07        |
> > |         | SalUn       |  5.11  |       15.09      |  98.41 |  92.75 |        1.25        |
> >
> > **Response to experimental details:**
> > Thank you for the suggestion. We provide more details about the experimental setup on the random forgetting and class wise forgetting scenarios here. We consider {andom forgetting and class-wise forgetting as our primary setups consistent with extant works. In the random forgetting scenario, we designate a fixed proportion of the number of samples from the training set (e.g. 10\%) to be set for deletion and the class-wise scenario considers the deletion of all samples from a certain class, signifying the deletion of a "concept". The experimental setup is mentioned in the supplemental material under section A.2. The methods utilize the ResNet-18 backbone and we additionally incorporate the ViT-Tiny over the SVHN dataset to broaden the setup and applicability. Unlearning is conducted with a $50\%$ saliency-based sparsity.
> >
> > Thank you for your suggestions, and we will add the above results and discussions to our revised version. Thank you!

---

### Review · Reviewer_bujA · 2025-10-03

**Summary Of Contributions:**

This work makes a number of contributions to the machine unlearning literature, which can be summarized as:

a. This work presents the theoretical analysis of FT methods for machine unlearning within a linear regression framework.

b. The analysis reveals that while FT models can achieve zero remaining loss, they fail to forget the forgetting data, as the pretrained model retains its influence and the fine-tuning process does not adequately mitigate it.

c. Building on this understanding, the paper introduces a Retention-Based Masking (RBM) strategy, which constructs a weight saliency map based on the remaining dataset, unlike existing methods that focus on the forgetting dataset.

d. This paper provides experiments on synthetic and real-world datasets validate the theoretical insights.

**Audience:**

Yes

**Audience Explanation:**

The work addresses a very timely and important problem in machine unlearning, which is directly connected to privacy regulations such as the right to be forgotten under GDPR. The paper also contributes a theoretical explanation of why fine-tuning fails to fully unlearn.

**Claims And Evidence:**

Yes

**Claims Explanation:**

The claims made in the submission are supported by accurate, convincing, and clear evidence. The paper provides:

a. Theoretical analysis of fine-tuning in approximate unlearning, with formal assumptions and theorems (e.g., Theorem 3.2, Theorem 3.4, Theorem 4.1).

b. Synthetic experiments that validate the theoretical results under both distinct and overlapping feature assumptions.

c. Comprehensive experiments on CIFAR-10 and CIFAR-100 demonstrate the effectiveness of the proposed Retention-Based Masking (RBM) method across multiple evaluation metrics.

**Requested Changes:**

a. The distinct/overlapping feature assumptions (Assumptions 3.1 and 3.3) are central, but the realism of these assumptions in high-dimensional nonlinear models is not sufficiently discussed. A clearer justification of how these cases map to practical settings is essential.

b. It is necessary to provide a more explicit computational complexity analysis of RBM.

c. The paper briefly mentions catastrophic forgetting. It will be more interesting to provide a more explicit connection to continual learning theory.

d. The t-SNE plots (Appendix A.2) could be brought into the main text to better illustrate the intuition.

---

> ### Author Response · Authors · 2025-10-13
> **Response**
>
> We are grateful to Reviewer bujA for the detailed review and positive feedback, and hope to address your concerns as follows:
>
> **Response to assumptions and models:** We thank the reviewer for the insightful comment regarding the feature assumptions, and we would like to provide further clarification here. Our feature assumptions are grounded in the theoretical setup of our model—a high-dimensional linear regression classifier—where distinct features refer to those unique to either the forgetting or retaining dataset, and overlapping features are shared between the two.
>
> For illustration, consider two categories: bananas and cars. Bananas have a distinct feature such as an elongated shape, while cars have a unique feature like mirrors; these are distinct features. Meanwhile, both may share overlapping features such as color—for example, both can be yellow.
>
> When extending this intuition to nonlinear models, one potential way is through a feature learning framework, where features can be identified at the patch level in high-dimensional representations. For instance, a class like "banana" can be represented as $x_{\text {banana }}=\left[y_1 \cdot u_{d_1}, y_1 \cdot u_o\right]$
> and "car" as
> $x_{\text {car }}=\left[y_2 \cdot u_{d_2}, y_2 \cdot u_o\right],$
> where $y_1, y_2$ denote the corresponding class labels, $u_{d_i}$ represent the distinct feature components, and $u_o$ corresponds to the overlapping feature component.
>
> Under this data model, the same analytical framework naturally extends to high-dimensional nonlinear networks, allowing our theoretical conclusions to generalize beyond the linear setting.
>
> **Response to complexity:** In the linear regression setting, Retention-Based Masking (RBM) involves three lightweight steps: (i) computing the gradient on the remaining dataset to estimate saliency, $O(N_r d)$; (ii) constructing the binary mask via a threshold $\gamma$, where $m_r=\mathbf{1}\{|\nabla_w L(w ; D_r)|_{w=w_o} \mid \geq \gamma\}$, which can be implemented either by applying a fixed threshold $(O(d))$; and (iii) applying the mask during the model update $(O(d^2))$. Overall, RBM maintains the same asymptotic complexity as fine-tuning, $O(N_r d+d^2)$.
>
> **Response to forgetting:** Thank you for your interest in our connection to continual learning theory. In continual learning, the main research focus is on mitigating catastrophic forgetting—for example, if the first task is to classify bananas and the second task is to classify cars, the goal is for the model to still correctly classify bananas after learning to classify cars. However, prior studies have shown that models often dramatically forget previously learned knowledge after fine-tuning on new tasks. This observation seems contradictory to our theoretical result in Theorem 3.2, which shows that a model can still retain information about the forgotten data even after fine-tuning solely on the remaining dataset. This apparent discrepancy may arise because continual learning fine-tunes on a new, unseen task and evaluates on previous tasks, whereas machine unlearning fine-tunes on a subset of previously seen data and evaluates on another subset of the same data.
>
> **Response to format:** Thank you for the suggestion. We will move the t-SNE plots from Appendix A.2 to the main text to better illustrate the intuition in the revised version.

---

### Decision · Action_Editor_7GVG · 2025-11-15

**Recommendation:** Accept with minor revision

**Additional Comments:**

This paper investigates why finetuning a pretrained model on the "retain set" (the part of the training set that we don't want to unlearn) can fail to forget a targeted "forget set" (the complement of the retain set in the training dataset).
The authors propose a theoretical framework for this, in the context of overparameterized linear regression, and use it to explain failure mode of finetuning for unlearning. The theoretical results consider two settings: one where the retain and forget sets have distinct features, and one where they have overlapping features.
They also propose a method called Retention-Based Masking (RBM), inspired by their analysis and show that RBM can facilitate unlearning, particularly in scenarios with overlapping features between the forgetting and remaining datasets. They also empirically examine the effectiveness of RBM on both synthetic and natural datasets, with results showing improvements some chosen baselines.

The reviewers appreciated the theoretical analysis, and found that the proposed method is a logical consequence of it. These results will be of interest to the TMLR community.
The reviewers also pointed out some weaknesses, including that (i) the theoretical analysis is confined to the overparameterized linear regression setting and assumptions made do not correspond to realistic scenarios; (ii) the empirical investigation includes only older baselines rather than state-of-the-art algorithms; (iii) limited datasets are used.
While I agree with the reviewers that the paper would be strengthened by improving on these points, and the authors partially addressed some of this feedback in their rebuttal (showing results on tini-imagenet and adding discussions for how theory could potentially be expanded), these do not stand in the way of acceptance to TMLR, so long as the claims are phrased in a way that accurately reflects the available evidence, which is mostly the case, as discussed, modulo the required changes that are listed below.

Changes required for minor revision

- Regarding the claim "We provide the first theoretical analysis of FT methods in the context of machine unlearning within a linear regression framework." Please comment on the relationship between the present paper and prior theoretical works that also study finetuning algorithms for unlearning. For example, Neel et al; Sekhari et al, which are already cited, but importantly also "Langevin Unlearning: A New Perspective of Noisy Gradient Descent for Machine Unlearning" by Chien et al (NeurIPS 2024). Please refine the claim of this being the "first theoretical analysis of FT [...]" as needed, or explain why this claim holds by discussing how this prior work differs. A comprehensive discussion of this should be added, e.g in the related work section.

- Regarding the claim that RBM is a novel masking strategy that "constructs a weight saliency map based on the remaining dataset, unlike existing methods that focus on the forgetting dataset", please discuss more recent work than Fan et al. for how saliency masks are constructed. Specifically, the following two more recent works should be cited here: (A) "Fast Machine Unlearning Without Retraining Through Selective Synaptic Dampening". Foster et al. AAAI 2024 and (B) "Improved Localized Machine Unlearning Through the Lens of Memorization". Torkzadehmahani et al. TMLR. For example, (A) uses both the forget and retain sets to inform the saliency mask, choosing to apply unlearning only on parameters that are disproportionately more important for the forget set relative to the retain set (see discussion in their paper). Please refine the claim that RBM is the first method to also use the retain set for the masking strategy accordingly, or explain in detail in related work why this is the case, in the context of these other works.

- Similarly, claims like "Our experimental results demonstrate that RBM achieves a better balance between unlearning and retaining objectives compared to existing masking approaches." aren't fully accurate as recent masking approaches have not been compared against, please make the claim more specific and accurate.

- [Optional but encouraged] Add a limitation section including unaddressed weaknesses and future work.

**Audience:**

Yes

**Audience Explanation:**

The reviewers consider this work of interest to the community, due to providing "a much-needed theoretical foundation for understanding fine-tuning" (Reviewer GWBK) and containing insights that are "valuable not only for researchers in privacy-preserving machine learning but also for those working on reliable and efficient model update strategies." (Reviewer GuEV).

**Claims And Evidence:**

Yes

**Claims Explanation:**

This paper investigates why finetuning a pretrained model on the "retain set" (the part of the training set that we don't want to unlearn) can fail to forget a targeted "forget set" (the complement of the retain set in the training dataset).
The authors propose a theoretical framework for this, in the context of overparameterized linear regression, and use it to explain failure mode of finetuning for unlearning. The theoretical results consider two settings: one where the retain and forget sets have distinct features, and one where they have overlapping features.
They also propose a method called Retention-Based Masking (RBM), inspired by their analysis and show that RBM can facilitate unlearning, particularly in scenarios with overlapping features between the forgetting and remaining datasets. They also empirically examine the effectiveness of RBM on both synthetic and natural datasets, with results showing improvements some chosen baselines.

The reviewers found that the claims are supported by accurate, convincing and clear evidence: "the evidence is convincing and clearly presented" (Reviewer GuEV); "the insight that masking should be based on the remaining data to preserve shared features is novel and well-supported by the theoretical results" (Reviewer GWBK); "The paper includes experiments on both synthetic and real-world datasets (CIFAR-10) to validate the theoretical claims. The synthetic experiments are particularly effective in illustrating the theoretical concepts of distinct and overlapping features." (Reviewer GWBK).

However, I have some concerns with some claims which I will ask the authors to incorporate in their final revision (see comment section below).

---

> ### Author Response · Authors · 2025-11-22
>
> We thank the reviewers and Editors for the valuable suggestions. In the revised version, we have clarified the distinctions between our theoretical analysis and prior works, as well as the differences between our masking approach and other existing methods, in the Related Work section. Additionally, we have added a discussion section in the appendix outlining unaddressed limitations and directions for future research.